# WHICH ALGORITHMS HAVE
# TIGHT GENERALIZATION BOUNDS?

## ABSTRACT

We study which machine learning algorithms have tight generalization bounds in the overparameterized setting. Our results build on and extend the recent work of Gastpar et al. (2024).

First, we present conditions that preclude the existence of tight generalization bounds. Specifically, we show that algorithms that have certain inductive biases that cause them to be unstable do not admit tight generalization bounds. Next, we show that algorithms that are sufficiently stable do have tight generalization bounds. We conclude with a simple characterization that relates the existence of tight generalization bounds to the conditional variance of the algorithm's loss.

## 1 INTRODUCTION

Generalization bounds are at the heart of learning theory, and they play a central role in attempts to mathematically explain the behavior of contemporary supervised machine learning systems. A generalization bound is an upper bound of the form

$$L_{\mathcal{D}}(A(S)) \leq b, \tag{1}$$

where $A(S)$ is the hypothesis output by learning algorithm $A$ when executed with training set $S$, and $L_{\mathcal{D}}(\cdot)$ represents the loss with respect to the population distribution $\mathcal{D}$. The term $b$ is typically an expression of the form

$$b = L_S(A(S)) + c(S, A(S), \mathcal{H}), \tag{2}$$

where $L_S(\cdot)$ is the empirical loss, $\mathcal{H}$ is a hypothesis class, and $c(S, A(S), \mathcal{H})$ is a 'complexity' term, such as the VC dimension or a spectral norm, etc.

We say that a generalization bound is *valid* if for every population distribution $\mathcal{D}$, Eq. (1) holds with high probability; we say that a valid bound is *uniformly tight* (Definition 2.4) if for every population distribution, with high probability the difference between the two sides of Eq. (1) is small.

Bounding the loss using a generalization bound is quite different from using a validation set. Technically, a generalization bound does not use additional samples beyond the training set $S$. And while a validation set provides a single post-hoc measurement of the population loss after training is complete, a good generalization bound can provide insight into *why* a learning algorithm performs well, and can offer guidance for model selection and the development of new learning algorithms. For a generalization bound to be useful in this way, it is important that the bound be tight, so that it can distinguish cases with small population loss from cases with larger loss.

Unfortunately, experimental works have shown that many of the generalization bounds of the form of Eq. (2) that have been proposed in the literature are vacuous[1] when applied to contemporary learning algorithms such as deep neural networks (Jiang et al., 2020; Dziugaite et al., 2020; Viallard et al., 2024, Section 4.4).

---

[1]A bound is *vacuous* if it is of the form $\mathbb{P}[L_{\mathcal{D}}(A(S)) \leq b] \geq 1 - \delta$ where $\delta \geq 1$ or (for the 0-1 loss) $b \geq 1$. Namely, it is a true statement that provides no guarantees on the performance of the algorithm.

Gastpar, Nachum, Shafer, and Weinberger (2024) offered a partial theoretical explanation for this empirical finding. They considered generalization bound as in Eq. (2), namely, bounds that depend only on the training set, the selected hypothesis, and the hypothesis class. They proved that any such bound cannot be uniformly tight in the overparameterized setting.[2] Therefore, they recommended focusing on generalization bounds involving expressions of the form $c(S, A(S), \mathcal{H}, A, \mathbb{D})$, i.e., bounds that depend also on the specific training algorithm and a specific collection $\mathbb{D}$ of population distributions for which the bound is intended.

This recommendation raises the following natural question:

> **Question 1.1.** *For which algorithms and distribution collections do there exist tight generalization bounds?*

This question was addressed in Theorems 3, 4 and 5 of Gastpar et al. (2024), but the general case remains open. In this paper we continue investigating this question, and present conditions that are necessary, sufficient, or necessary and sufficient for the existence of tight generalization bounds for a given learning algorithm and distribution collection.

## 1.1 SETTING

Following Gastpar et al. (2024), we study the existence of tight generalization bounds using a notion of *estimability*.

**Definition 1.2** (Estimability). *Let $\mathcal{X}$ and $\mathcal{Y}$ be sets, let $m \in \mathbb{N}$, let*

$$A: \ (\mathcal{X} \times \mathcal{Y})^m \to \mathcal{Y}^{\mathcal{X}}$$

*be a learning rule, and let $\mathbb{D} \subseteq \Delta(\mathcal{X} \times \mathcal{Y})$ be a collection of distributions. An $\underline{estimator}$ is a function*

$$\mathcal{E}: \ (\mathcal{X} \times \mathcal{Y})^m \to \mathbb{R}.$$

*Let $\varepsilon, \delta \in [0, 1]$. We say that $A$ is $\underline{uniformly\ estimable\ (or\ worst\text{-}case\ estimable)}$ with respect to distributions $\mathbb{D}$ with precision $\underline{\varepsilon}$ and confidence $\delta$ using $m$ samples if there exists an estimator $\mathcal{E}$ such that*

$$\forall \mathcal{D} \in \mathbb{D}: \ \mathbb{P}_{S \sim \mathcal{D}^m}\big[\big|\mathcal{E}(S) - L_{\mathcal{D}}(A(S))\big| \le \varepsilon\big] \ge 1 - \delta.$$

*We say that $A$ is $\underline{estimable\ on\ average}$ with respect to distributions $\mathbb{D}$ with precision $\varepsilon$ and confidence $\delta$ using $m$ samples if there exists an estimator $\mathcal{E}$ such that*

$$\mathbb{P}_{\mathcal{D} \sim \mathrm{U}(\mathbb{D}), S \sim \mathcal{D}^m}\big[\big|\mathcal{E}(S) - L_{\mathcal{D}}(A(S))\big| \le \varepsilon\big] \ge 1 - \delta.$$

*(More briefly, we say that $(A, \mathbb{D})$ is $(\varepsilon, \delta, m)$-uniformly estimable, or $(\varepsilon, \delta, m)$-estimable on average.)*

The connection between estimability and tight generalization bounds is as follows.

**Fact 1.3.** *Using the notation of Definition 1.2, if $(A, \mathbb{D})$ is $(\varepsilon, \delta, m)$-estimable on average, then there exists a generalization bound $b(S)$ (that may depend on $A$ and $\mathbb{D}$) that is $\varepsilon$-tight on average, namely*

$$\mathbb{P}_{\mathcal{D} \sim \mathrm{U}(\mathbb{D}), S \sim \mathcal{D}^m}[b(S) - \varepsilon \le L_{\mathcal{D}}(A(S)) \le b(S)] \ge 1 - \delta. \tag{3}$$

*Indeed, the generalization bound is simply $b(S) = \mathcal{E}(S) + \varepsilon$, where $\mathcal{E}$ is the estimator witnessing the estimability of $(A, \mathbb{D})$.*

*In the other direction, if $(A, \mathbb{D})$ is not $(\varepsilon, \delta, m)$-estimable on average, then there exists no generalization bound that satisfies Eq. (3), and in particular no generalization bound can be uniformly tight (as in Definition 2.4).*

The main question studied in this paper is as follows: which general and useful conditions are necessary, sufficient, or necessary and sufficient for a tuple $(A, \mathbb{D})$ to be $(\varepsilon, \delta, m)$-uniformly estimable, or $(\varepsilon, \delta, m)$-estimable on average?

---

[2] They actually showed a stronger result, that such bounds are not tight in an average-case sense for many (algorithms, distribution) pairs.

We are specifically interested in addressing these questions in settings where the number of samples is not sufficient to guarantee learning in general (in the sense of the VC theorem for example), because most contemporary machine learning algorithms (such as deep neural networks) are used in such settings. This is captured by the following definition.[3]

**Definition 1.4** (Overparameterized setting). *Let $\mathcal{X}$ and $\mathcal{Y}$ be sets, let $\mathbb{D} \subseteq \Delta(\mathcal{X} \times \mathcal{Y})$, let $\alpha, \beta \in [0,1]$ and $m \in \mathbb{N}$. We say that $(\mathbb{D}, m)$ is $\underline{(\alpha, \beta)\text{-learnable}}$ if there exists a learning rule $A : (\mathcal{X} \times \mathcal{Y})^m \to \mathcal{Y}^{\mathcal{X}}$ such that*

$$\mathbb{P}_{\mathcal{D} \sim \mathrm{U}(\mathbb{D}), S \sim (\mathcal{D})^m}[L_{\mathcal{D}}(A(S)) \leq \alpha] \geq 1 - \beta.$$

*We say that $(\mathbb{D}, m)$ is $\underline{(\alpha, \beta)\text{-overparameterized}}$ if it is not $(\alpha, \beta)$-learnable.*

### 1.2 EXAMPLES

We present a few simple examples to showcase the richness of the estimability setting. In this section $\varepsilon, \delta \in (0,1)$, $\mathcal{X}$ is a set, $m \in \mathbb{N}$ is a sample size, $A : (\mathcal{X} \times \{\pm 1\})^m \to \{\pm 1\}^{\mathcal{X}}$ is a learning rule, and $S = ((x_1, y_1), \ldots, (x_m, y_m))$ is a training set.

**Example 1.5** (Perfect learnability does not imply perfect estimability). Let $\mathcal{X} = [0,1]$, let $\mathbb{D} = \Delta(\mathcal{X} \times \{1\})$ be the set of all distributions of labeled examples $(x, y)$ where $x \in \mathcal{X}$ and $y = 1$. The collection $\mathbb{D}$ is perfectly learnable, that is, there exists a learning algorithm that always achieves 0 population loss (namely, the learning algorithm that always outputs the constant function $h(x) = 1$).

Nonetheless, not every learning algorithm is worst-case estimable with respect to $\mathbb{D}$. Indeed, consider the algorithm $A$ that on input $S$ outputs the hypothesis

$$h(x) = \begin{cases} -1 & x \in \{x_1, \ldots, x_m\} \\ +1 & \text{otherwise.} \end{cases}$$

For any distribution $\mathcal{D} \in \mathbb{D}$, $L_{\mathcal{D}}(A(S)) = \mathcal{D}_{\mathcal{X}}(\{x_1, \ldots, x_m\})$, where $\mathcal{D}_{\mathcal{X}}$ is the marginal of $\mathcal{D}$ on $\mathcal{X}$. Hence, estimating the loss of $A$ is equivalent to a task of support size estimation, which is difficult. Concretely, for any finite set $T \subseteq \mathcal{X}$, let $\mathcal{D}_T = \mathrm{U}(T \times \{1\})$. Let $\mathcal{E}$ be any estimator, and consider an experiment where with probability $1/2$, we sample $T \sim \mathrm{U}(\mathcal{X})^{m^2}$ and set $\mathcal{D} = \mathcal{D}_T$, and with probability $1/2$ we set $\mathcal{D} = \mathcal{D}_{\mathrm{U}} := \mathrm{U}(\mathcal{X} \times \{1\})$. Consider the probability

$$p = \mathbb{P}_{S \sim \mathcal{D}^m}\left[\left|\mathcal{E}(S) - L_{\mathcal{D}}(A(S))\right| \geq \frac{1}{2m}\right].$$

Let $E$ be the event where $|\{x_1, \ldots, x_m\}| = m$. In the case where $\mathcal{D} = \mathcal{D}_T$ with $|T| = m^2$, Claim L.2 implies that $\mathbb{P}[E] \geq 1/e$. And in the case where $\mathcal{D} = \mathcal{D}_{\mathrm{U}}$, $\mathbb{P}[E] = 1$. Hence, in both cases, with probability at least $1/e$, the estimator receives a sample of $m$ distinct points chosen independently and uniformly from $\mathcal{X}$, and it cannot distinguish between these two cases. However, $L_{\mathcal{D}_{\mathrm{U}}}(A(S)) = 0$, whereas $L_{\mathcal{D}_T}(A(S)) = \frac{1}{m}$ when $E$ occurs. This implies that $p \geq 1/2e$, and so $(A, \mathbb{D})$ is not $(\frac{1}{2m}, \delta, m)$-uniformly estimable for any $\delta < 1/2e$. $\qquad\square$

Some algorithms are very estimable but are not good learning algorithms, as in the following three examples.

**Example 1.6** (Constant algorithms are estimable). Let $m \geq \log(1/\delta)/\varepsilon^2$. Let $h_0 : \mathcal{X} \to \{\pm 1\}$ be a function, and let $A$ be the constant learning algorithm such that $A(S) = h_0$ for all $S$. Then by Hoeffding's inequality, $A$ is $(\varepsilon, \delta, m)$-uniformly estimable with respect to the set of all distributions $\mathbb{D} = \Delta(\mathcal{X} \times \{\pm 1\})$, with estimator $\mathcal{E}(S) = L_S(h_0)$. $\qquad\square$

For some algorithms, the empirical loss is not a good estimator, yet the algorithm is still estimable.

**Example 1.7** (Memorization). Let $\Omega(\log(1/\delta)/\varepsilon^2) \leq m \leq O(\varepsilon|\mathcal{X}|)$, and consider the algorithm $A$ that on input $S$, outputs the hypothesis

$$h(x) = \begin{cases} y & \exists y : \{y\} = \{y_i : i \in [m] \wedge x_i = x\} \\ -1 & \text{otherwise.} \end{cases}$$

---

[3]This is Definition 2 in Gastpar et al. (2024). See further discussion in Appendix B.

Let $\mathbb{D}$ be the collection of all distributions over $\mathcal{X} \times \{\pm 1\}$ that have a uniform marginal on $\mathcal{X}$. Note that $A$ always has 0 empirical loss. However, $(A, \mathbb{D})$ is $(\varepsilon, \delta)$-uniformly estimable, using $\mathcal{E}(S) = |\{i \in [m] : y_i = 1\}| / m$. $\qquad\square$

**Example 1.8** (Most algorithms are estimable). Let $d = |\mathcal{X}| < \infty$, let $\mathcal{F} = \{\pm 1\}^{\mathcal{X}}$, and for each $f \in \mathcal{F}$, let $\mathcal{D}_f = \mathrm{U}(\{(x, f(x)) : x \in \mathcal{X}\})$. Let $\mathcal{A}$ be the set of all mappings $(\mathcal{X} \times \{\pm 1\})^m \to \{\pm 1\}^{\mathcal{X}}$, and consider a mapping $A$ chosen uniformly from the set $\mathcal{A}$. For any fixed $f \in \mathcal{F}$ and for any fixed sample $S$ of size $m$ consistent with $f$, $A(S)$ is a function that was chosen uniformly from $\mathcal{F}$. By Hoeffding's inequality,

$$\forall f \in \mathcal{F} \; \forall S \in \mathrm{supp}(\mathcal{D}_f) : \; \mathbb{P}_{A \sim \mathrm{U}(\mathcal{A})}\left[\left|L_{\mathcal{D}_f}(A(S)) - \frac{1}{2}\right| \geq \varepsilon\right] \leq 2e^{-2d\varepsilon^2}.$$

In particular,

$$\mathbb{P}_{A \sim \mathrm{U}(\mathcal{A}), f \sim \mathrm{U}(\mathcal{F}), S \sim (\mathcal{D}_f)^m}\left[\left|L_{\mathcal{D}_f}(A(S)) - \frac{1}{2}\right| \geq \varepsilon\right] \leq 2e^{-2d\varepsilon^2}.$$

Hence, by Markov's inequality, 99% of algorithms $A \in \mathcal{A}$ satisfy that $(A, \{\mathcal{D}_f\}_{f \in \mathcal{F}})$ is $(\varepsilon, 200e^{-2d\varepsilon^2}, m)$-estimable on average. $\qquad\square$

A similar argument shows also that most ERM algorithms are estimable in the overparameterized setting.[4] In both cases, the algorithms are estimable because their loss is guaranteed to be high, namely, the algorithms are poor learners.

Finally, algorithms for learning parity functions are a particularly instructive case.

**Example 1.9** (Parity functions). Let $d \in \mathbb{N}$ be large enough, $\mathcal{X} = (\mathbb{F}_2)^d$, and let $\mathcal{H} = \{f_w : w \in \mathcal{X}\} \subseteq (\mathbb{F}_2)^{\mathcal{X}}$ be the class of parity functions such that $f_w(x) = \sum_{i \in [d]} w_i \cdot x_i$. Let $\mathbb{D} = \{\mathcal{D}_f\}_{f \in \mathcal{H}}$ with $\mathcal{D}_f = \mathrm{U}(\{(x, f(x)) : x \in \mathcal{X}\})$. For a learning rule $A$ and sample size $m$, let

$$p(m) = \mathbb{P}_{\substack{\mathcal{D} \sim \mathrm{U}(\mathbb{D}) \\ S \sim (\mathcal{D})^m}}[L_{\mathcal{D}}(A(S)) = 0].$$

For sample size $m \geq d + 10$, any ERM algorithm for $\mathcal{H}$ satisfies[5] $p(m) \geq 0.999$, meaning it learns $\mathbb{D}$ well, and hence is $(0, 10^{-3}, d + 10)$-estimable on average.

Similarly, for smaller sample sizes, any ERM for $\mathcal{H}$ satisfies $p(d) \geq 0.61$, and $p(d-1) \geq 0.38$. However, ERM algorithms differ in their degree of estimability for smaller sample sizes. Concretely, there exist ERM algorithms such that for any $6 \leq m \leq d$ there exists a collection $\mathbb{D}_m$ for which the algorithm is not $(0.25, 0.32, m)$-estimable on average. In contrast, for the same hard collections $\mathbb{D}_m$, ERM algorithms without an inductive bias perform poorly on all distributions for small $m$, so they are significantly more estimable. $\qquad\square$

ERM algorithms for parity functions demonstrate two important phenomena: (1) Estimability can be a very delicate matter, in the sense that changing the sample size by a small additive constant can make all the difference (e.g., any ERM for parities is very estimable with $m = d + 10$ samples, but not very estimable with $m = d$); (2) when the sample size is not sufficient for learning all the distributions in the collection $\mathbb{D}$, there can be a trade-off between learning performance and estimability. Algorithms with no inductive bias will perform equally poorly for all distributions, and this makes them estimable. In contrast, algorithms that have an inductive bias towards a subset $\mathbb{D}' \subseteq \mathbb{D}$ can perform well on $\mathbb{D}'$, and this can make them less estimable.

---

[4]Consider an overparameterized setting with $m = o(d)$. The output of any (realizable) ERM will have zero error on points in $S$, and will make an error on each unseen data points with probability $1/2$, yielding an expected population loss of $(d - m)/2d \approx 1/2$. Hence, essentially the same result as in Example 1.8 can be obtained also for ERMs by applying Hoeffding's inequality.

[5]The quantitative statements in this example follow from the results in Gastpar et al., 2024, see Appendix D for a discussion.

### 1.3 OUR RESULTS

We investigate which algorithms and collections of distributions are estimable. Recall that in Example 1.9 we saw that estimability is a delicate phenomenon. In particular, changing the sample size by just a small constant number can in some cases drastically change the set of $(\varepsilon, \delta)$ estimability parameters that are achievable. This means that identifying a simple and tight characterization that precisely determines the number of samples necessary and sufficient for estimability can be a difficult undertaking.

In this paper, we present conditions that preclude estimability, conditions that guarantee estimability, and a condition that is both necessary and sufficient for estimability.

Our first result is a condition that precludes estimability for algorithms that have an inductive bias towards certain subsets of VC classes, showing a connection between estimability and a central notion from traditional learning theory.

**Theorem** (Informal version of Theorem 3.1). *Let $\mathcal{H} \subseteq \{\pm 1\}^{\mathcal{X}}$ be a hypothesis class with VC dimension $d$ large enough, and let $m \leq \sqrt{d}/10$. Then there exists a subset $\mathcal{F} \subseteq \mathcal{H}$ and corresponding realizable distributions $\mathbb{D}$ such that any learning rule that has an inductive bias towards $\mathcal{F}$ is not $(1/4 - o(1), 1/6, m)$-estimable on average over $\mathbb{D}$.*[6]

Note that the theorem precludes estimability on average, and so in particular it precludes worst-case estimability. The proof of Theorem 3.1 uses the Johnson–Lindenstrauss lemma (Theorem L.1), the probabilistic method, and a technical lemma (Lemma I.1) concerning the estimability of nearly-orthogonal functions.

To the best of our knowledge, this paper is the first to provide a rigorous and general mathematical formulation showing that any finite VC class admits inestimable algorithms. This is somewhat surprising because it means, for instance, that for any neural network architecture, there are some training algorithms for which one will not be able to derive tight generalization bounds (even if the distribution is realizable!). We believe this is a meaningful contribution.

Our next inestimability result is as follows.

**Theorem** (Informal version of Theorem 3.2). *Let $\mathcal{H} \subseteq \{\pm 1\}^{\mathcal{X}}$ be a collection of roughly $2^m$ nearly-orthogonal functions and corresponding realizable distributions $\mathbb{D}$. Then any learning rule that has an inductive bias towards $\mathcal{H}$ is not $(1/4 - o(1), \sim 1/6, m)$-estimable on average over $\mathbb{D}$.*[7]

Theorem 3.2 is partially stronger than Theorem 3.1 in the sense that it shows inestimability for *every* algorithm that has an inductive bias towards a class of nearly-orthogonal functions, whereas Theorem 3.1 only shows the existence of a subclass with this property.[8] On the other hand, Theorem 3.1 is stronger than Theorem 3.2 in the sense that if Theorem 3.2 is applied to show inestimability for subclasses of a VC class, then it yields inestimability only for $m \leq O\left(\sqrt[3]{d}\right)$, whereas Theorem 3.1 obtains inestimability for all $m \leq O\left(\sqrt{d}\right)$.[9]

To show Theorem 3.2, we prove a concentration inequality using the duality of linear programs (Lemma J.1), and then invoke the technical lemma (Lemma I.1).

**Remark 1.10.** *Theorems 3.1 and 3.2 are stated for the case of binary labels, but they immediately imply inestimability also for regression and multi-class classification.*

---

[6]Note that $(\mathbb{D}, m)$ is roughly a $(1/4 - o(1), 1/6)$-overparameterized setting. See Remark 3.3.

[7]Similarly, $(\mathbb{D}, m)$ is roughly a $(1/4 - o(1), 0.24)$-overparameterized setting. See Remark 3.3.

[8]Additionally the quantity hidden by the $o(1)$ notation is smaller in Theorem 3.2 by a quadratic factor (order $1/m$ vs. $1/\sqrt{m}$).

[9]The limitation $m \leq O\left(\sqrt[3]{d}\right)$ when using Theorem 3.2 follows from the tightness of the Johnson–Lindenstrauss (JL) lemma. By the JL lemma, taking a collection $\mathcal{F}$ of $2^m$ orthogonal functions on a high dimensional domain, we can project $\mathcal{F}$ using a random projection and obtain a collection $\mathcal{F}'$ of $2^m$ functions that are $\varepsilon$-orthogonal defined on a domain of dimension $\log(2^m)/\varepsilon^2$. In particular, let $\mathcal{H}$ be a class with VC dimension $d$. We want to project $\mathcal{F}$ onto an $\mathcal{H}$-shattered set of size $d$ with $\varepsilon = \Theta(1/m)$. This yields $d = m/(\Theta(1/m))^2 = \Theta(m^3)$. The tightness of JL implies that this construction cannot be improved.

One way to interpret Theorems 3.1 and 3.2 is to consider a scenario where one derives a new generalization bound for a given algorithm, without making explicit distributional assumptions (as is the case for many published generalization bounds), and having a sample size within the regime of our theorems. Such bounds are generally formulated as high probability upper bounds on the population loss. Note that the lack of distributional assumptions means that the bound has to hold (be a valid upper bound) for all distributions, including the families of distributions that appear in our theorems.

But this means, in the light of our theorems, that the considered bound is necessarily very weak for many distributions unless one satisfies at least one of the following items:

1. Exclude in advance all families of distributions with nearly-orthogonal labeling functions, and use this fact in the derivation of the generalization bound.

2. Mathematically show that the algorithm is not biased towards any set of nearly-orthogonal functions.[10]

The intuition behind Theorem 3.2 is that having an inductive bias towards a collection $\mathcal{H}$ of nearly-orthogonal functions makes the algorithm very unstable – small changes in the training set will cause the algorithm to shift between hypotheses in $\mathcal{H}$, which are all very different from one another. This motivates our next result, which shows that stable algorithms are estimable, as follows.

**Theorem** (Informal version of Theorem 4.3). *Let A be an algorithm that is sufficiently stable with respect to a collection of distributions $\mathbb{D}$ (in a sense of loss stability or hypothesis stability similar to Rogers and Wagner, 1978, or Kearns and Ron, 1999). Then $(A, \mathbb{D})$ is estimable.*

Seeing as there are many definitions of stability in the literature, Theorem 4.3 makes a nontrivial conceptual contribution by identifying the "correct" notion of stability for understanding estimability. Other notions of stability, such as leave-one-out stability (Bousquet & Elisseeff, 2002), do not capture estimability as well, as we discuss in Section 4.

An additional motivation for Theorem 4.3 is the intuition that contemporary machine learning algorithms (like deep neural networks) might indeed be sufficiently stable. If so, Theorem 4.3 would apply, meaning that it is possible to obtain tight generalization bounds for deep neural networks based on the stability property. To substantiate this intuition, we conduct simple preliminary experiments to estimate the the stability of neural networks in practice. Our empirical findings, presented in Appendix M, suggest that neural networks are indeed quite stable.

Finally, in Section 5, we present a necessary and sufficient condition for estimability based on the conditional variance of the algorithm's loss. This characterization is formalized in terms of $\ell_2$ estimability, which is asymptotically equivalent to average case estimability via Markov's inequality.

**Fact** (Fact 5.2). *A is $(\varepsilon, m)$-estimable in $\ell_2$ with respect to $\mathbb{D}$ if and only if*

$$\mathbb{E}[\text{var}(L_{\mathcal{D}}(A(S)) \mid S)] \leq \varepsilon.$$

### 1.4 Related Works

The most closely related works to our study are those by Gastpar et al. and prior research on stability, which we will examine in detail in this section. For a broader comparison to other studies addressing generalization bounds and their limitations, we refer the reader to Appendix A.

#### 1.4.1 Comparison to Gastpar et al. (2024)

The estimability setting studied in our paper was introduced by Gastpar, Nachum, Shafer, and Weinberger (2024). In Theorem 3 of their paper, they show a limitation on estimability (a learnability–estimability trade-off) for algorithm-dependent bounds that is fairly abstract

---

[10]It is known that there exist at least some neural network architectures which, when trained with SGD, are capable of learning orthogonal functions (such as parities). See Theorem 1 in Abbe and Sandon (2020).

and involves a total variation condition that might be hard to check in many cases. In contrast, Theorems 3.1 and 3.2 involve very concrete combinatorial and geometric conditions (VC dimension, orthogonal functions). Theorems 4 and 5 in their paper are more concrete, but they hold only for exactly orthogonal functions with strict algebraic structure (parity functions). In contrast, our Theorem 3.2 applies generally to any nearly-orthogonal function class (including classes that are exactly-orthogonal as a special case).

Unlike Gastpar et al. (2024), our work also presents positive results (Theorem 4.3 and Fact 5.2), showing cases where generalization bounds for specific algorithms can be tight even in the overparameterized setting. The conceptual connections between estimability, stability and conditional variance appearing in those results was not present in Gastpar et al. (2024).

Finally, our techniques also differ from those of Gastpar et al. (2024). We use the Johnson–Lindenstrauss lemma, our technical lemma (Lemma I.1), and the duality of linear programming — expanding the arsenal of tools readily available for the study of estimability.

In summary, our work builds upon the foundation laid by Gastpar et al. (2024), but we make several important contributions that go beyond their results.

### 1.4.2 STABILITY

In Definitions 4.1 and 4.2, we formalize simple stability conditions that guarantee the existence of tight generalization bounds, as we show in Theorem 4.3. There are many definitions of stability in the literature, and it is important to appreciate that Theorem 4.3 makes a nontrivial conceptual contribution by identifying the "correct" notion of stability for understanding estimability.

Definitions 4.1 and 4.2 are similar to the definition of hypothesis stability and loss stability in Kearns and Ron (1999), Elisseeff, Evgeniou, and Pontil (2005), and Rogers and Wagner (1978). Lei, Jin, and Ying (2022) use another similar definition for stability and utilize it to derive generalization bounds for GD and SGD.

In contrast, our definitions of stability are also reminiscent of the replace-one stability in Bousquet and Elisseeff (2002), but as we explain in Section 4, our definitions overcome an important limitation present in their definition. In particular, the memorization algorithm (Example 1.7), which is very estimable, is not stable according to the definition of stability of Bousquet and Elisseeff (2002), but it is stable according to our definitions.

## 2 PRELIMINARIES

**Definition 2.1.** *For $m \in \mathbb{N}$ and sets $\mathcal{X}$ and $\mathcal{Y}$, a learning rule is a function $A : (\mathcal{X} \times \mathcal{Y})^m \to \mathcal{Y}^{\mathcal{X}}$. We will also consider learning rules with variable-size input, i.e., $A : (\mathcal{X} \times \mathcal{Y})^* \to \mathcal{Y}^{\mathcal{X}}$.*

In this paper we informally use the terms 'learning algorithm' and 'learning rule' interchangeably. Both words refer to a function, ignoring considerations of computability. All learning algorithms in this paper are deterministic.[11]

**Notation 2.2.** *For a set $\Omega$, we write $\Delta(\Omega)$ to denote the collection of all probability measures over a measurable space $(\Omega, \mathcal{F})$, where $\mathcal{F}$ is some fixed $\sigma$-algebra that is implicitly understood. We write $\mathrm{U}(\Omega)$ to denote the uniform distribution over $\Omega$.*

**Definition 2.3.** *Let $m \in \mathbb{N}$, let $\mathcal{X}$, $\mathcal{Y}$ be sets, let $h : \mathcal{X} \to \mathcal{Y}$, let $S = ((x_1, y_1), \ldots, (x_m, y_m)) \in (\mathcal{X} \times \mathcal{Y})^m$, and let $\mathcal{D} \in \Delta(\mathcal{X} \times \mathcal{Y})$. The empirical loss of $h$ with respect to $S$ is $L_S(h) = \frac{1}{m} \sum_{i \in [m]} \mathbb{1}(h(x_i) \neq y_i)$. The population loss of $h$ with respect to $\mathcal{D}$ is $L_{\mathcal{D}}(h) = \mathbb{P}_{(x,y) \sim \mathcal{D}}[h(x) \neq y]$.*

**Definition 2.4** (Uniformly tight generalization bound for an algorithm)**.** *Let $m \in \mathbb{N}$, $\varepsilon, \delta \in [0, 1]$, let $\mathcal{X}$ and $\mathcal{Y}$ be sets, let $m \in \mathbb{N}$, let $A : (\mathcal{X} \times \mathcal{Y})^m \to \mathcal{Y}^{\mathcal{X}}$ be a learning rule, and let $b : (\mathcal{X} \times \mathcal{Y})^m \to [0, 1]$ be a generalization bound (that may depend on $A$). We say that $b$ is uniformly tight for $A$ with precision $\varepsilon$ and confidence $\delta$ if for any distribution $\mathcal{D} \in \Delta(\mathcal{X} \times \mathcal{Y})$,*

$$\mathbb{P}_{S \sim \mathcal{D}^m}[b(S) - \varepsilon \leq L_{\mathcal{D}}(A(S)) \leq b(S)] \geq 1 - \delta.$$

---

[11]See Appendix C for a discussion on how our results can be extended to randomized algorithms.

**Notation 2.5.** *Let $\mathcal{X}$ be a set, let $\mathcal{F} \subseteq \{\pm 1\}^{\mathcal{X}}$ be a hypothesis class, and let $S \in (\mathcal{X} \times \{\pm 1\})^*$. We denote $\mathcal{F}_S = \{f \in \mathcal{F} : L_S(f) = 0\}$.*

The following definition captures the notion of a learning rule having an inductive bias towards a particular set of hypotheses.

**Definition 2.6.** *Let $m \in \mathbb{N}$, let $\mathcal{X}$ be a set, and let $\mathcal{F} \subseteq \{\pm 1\}^{\mathcal{X}}$ be a hypothesis class. We say that a learning rule $A : (\mathcal{X} \times \{\pm 1\})^m \to \{\pm 1\}^{\mathcal{X}}$ is $\underline{\mathcal{F}\text{-interpolating}}$ if $A(S) \in \mathcal{F}_S$ for every sample $S \in (\mathcal{X} \times \{\pm 1\})^m$ such that $\mathcal{F}_S \neq \varnothing$.*

**Remark 2.7.** *The property of $\mathcal{F}$-interpolation is similar to the more common property of proper empirical risk minimization (proper ERM) for $\mathcal{F}$. However, $\mathcal{F}$-interpolation is a slightly weaker requirement. Specifically, if $S$ is not $\mathcal{F}$-realizable (i.e., $\mathcal{F}_S = \varnothing$), then an $\mathcal{F}$-interpolating learning rule may output any function in $\{\pm 1\}^{\mathcal{X}}$, whereas a proper learning rule for $\mathcal{F}$ must always output a function from $\mathcal{F}$.*

**Definition 2.8.** *Let $\varepsilon \geq 0$, let $\mathcal{X}$ be a set, and let $\mathcal{F} \subseteq \{\pm 1\}^{\mathcal{X}}$ be a hypothesis class. We say that $\mathcal{F}$ is $\underline{\varepsilon\text{-orthogonal with respect to } \mathcal{X}}$, denoted $\mathcal{F} \in \perp_{\varepsilon, \mathcal{X}}$, if every distinct $f, g \in \mathcal{F}$ satisfy*

$$\left| \mathbb{E}_{x \sim \mathrm{U}(\mathcal{X})}[f(x)g(x)] \right| \leq \varepsilon.$$

*For simplicity, we write $\mathcal{F} \in \perp_{\varepsilon}$ when $\mathcal{X}$ is understood from context.*

**Fact 2.9.** *Let $\varepsilon > 0$ and let $\mathcal{F} \subseteq \{\pm 1\}^{\mathcal{X}}$ be $\varepsilon$-orthogonal. Then for any distinct $f, g \in \mathcal{F}$,*

$$\frac{1}{2} - \frac{\varepsilon}{2} \leq \mathbb{P}_{x \sim \mathrm{U}(\mathcal{X})}[f(x) = g(x)] \leq \frac{1}{2} + \frac{\varepsilon}{2}.$$

*Proof.*     $\mathbb{P}_{x \sim \mathrm{U}(\mathcal{X})}[f(x) = g(x)] = \mathbb{E}_{x \sim \mathrm{U}(\mathcal{X})}[\mathbb{1}(f(x) = g(x))] = \mathbb{E}_{x \sim \mathrm{U}(\mathcal{X})}\left[ \frac{1 + f(x)g(x)}{2} \right]$

$$= \frac{1}{2} + \frac{1}{2} \cdot \mathbb{E}_{x \sim \mathrm{U}(\mathcal{X})}[f(x)g(x)]. \qquad \square$$

## 3    Conditions that Preclude Estimability

We present two conditions that preclude estimability.

### 3.1    Inestimability for VC Classes

**Theorem 3.1.** *There exists $d_0 > 0$ as follows. For any integer $d \geq d_0$, let $\mathcal{X}$ be a set, let $\mathcal{H} \subseteq \{\pm 1\}^{\mathcal{X}}$ such that $\mathsf{VC}(\mathcal{H}) = d$, and let $m \in \mathbb{N}$ such that $m \leq \sqrt{d}/10$. Then there exists a subset $\mathcal{F} \subseteq \mathcal{H}$ and a collection $\mathbb{D} \subseteq \Delta(\mathcal{X} \times \{\pm 1\})$ of $\mathcal{F}$-realizable distributions such that for any $\mathcal{F}$-interpolating learning rule $A$ and for any estimator $\mathcal{E} : (\mathcal{X} \times \{\pm 1\})^m \to [0, 1]$ that may depend on $\mathbb{D}$ and $A$,*

$$\mathbb{P}_{\substack{\mathcal{D} \sim \mathrm{U}(\mathbb{D}) \\ S \sim \mathcal{D}^m}}\left[ \left| \mathcal{E}(S) - L_{\mathcal{D}}(A(S)) \right| \geq \frac{1}{4} - \frac{1}{2d^{1/4}} \right] \geq \frac{1}{6}. \tag{4}$$

The proof of Theorem 3.1 appears in Appendix E. We note that some of the constants appearing in the theorem were chosen for simplicity, and can be improved.

### 3.2    Inestimability for Nearly-Orthogonal Functions

**Theorem 3.2.** *Let $m \in \mathbb{N}$, let $\mathcal{X}$ be a set, and let $A : (\mathcal{X} \times \{\pm 1\})^m \to \{\pm 1\}^{\mathcal{X}}$ be a learning rule. Assume that $A$ is $\mathcal{F}$-interpolating for a set $\mathcal{F} \subseteq \{\pm 1\}^{\mathcal{X}'}$ where $\mathcal{X}' \subseteq \mathcal{X}$, $100m^2 \leq |\mathcal{X}'| < \infty$, $\mathcal{F} \in \perp_{1/1000m, \mathcal{X}'}$ and $|\mathcal{F}| = 2^m + 1$. Then there exists a collection of $\mathcal{F}$-realizable distributions $\mathbb{D} \subseteq \Delta(\mathcal{X}' \times \{\pm 1\})$ such that for any estimator function $\mathcal{E} : (\mathcal{X} \times \{\pm 1\})^m \to [0, 1]$ that may depend on $\mathbb{D}$ and $A$,*

$$\mathbb{P}_{\substack{\mathcal{D} \sim \mathrm{U}(\mathbb{D}) \\ S \sim \mathcal{D}^m}}\left[ \left| \mathcal{E}(S) - L_{\mathcal{D}}(A(S)) \right| \geq \frac{1}{4} - \frac{1}{4000m} \right] \geq 0.16.$$

The proof of Theorem 3.2 appears in Appendix F. We note that here too, the constants appearing in the theorem were chosen for simplicity, and can be improved. In particular, using a similar technique it is possible to show a lower bound of 1/6 instead of 0.16, matching the bound in Theorem 3.1.

**Remark 3.3.** *Both theorems in Section 3 show inestimability results in the overparameterized setting (Definition 1.4).[12] In particular, examining Eq. (7) in the proof of Theorem 3.1 reveals that the pair $(\mathbb{D}, m)$ appearing in the theorem statement constitutes a*

$$\left( \frac{1}{4} - \frac{1}{4d^{1/4}} - \xi, \ \frac{1}{6} - \xi' \right)$$

*overparameterized setting for any non-negative $\xi$ and $\xi'$ where at least one is positive.*

*Similarly, Eq. (12) in the proof of Theorem 3.2 implies that $(\mathbb{D}, m)$ in that theorem is a*

$$\left( \frac{1}{4} - \frac{1}{4000m} - \xi, \ 0.24 - \xi' \right)$$

*overparameterized setting for $\xi$ and $\xi'$ as above.[13]*

## 4 Sufficient Conditions for Estimability

In Examples 1.6 and 1.7 we saw that the constant algorithm and the memorization algorithm are very estimable. These algorithms are also very stable. Indeed, they always output the same (or essentially the same) hypothesis.[14] In the other direction, Theorem 3.2 shows that certain algorithms that are very unstable, are not estimable. This suggests that stability might play an important role in determining the estimability of an algorithm.

One notion of algorithmic stability that is common in the literature is leave-one-out stability (Bousquet & Elisseeff, 2002). However, it is easy to see that the memorization algorithm, which is estimable and is (intuitively) very stable, does not satisfy their definition of stability. Therefore, we use the following alternative definitions of algorithmic stability, which are similar to Rogers and Wagner (1978) and Kearns and Ron (1999).

**Definition 4.1.** *Let $m, k \in \mathbb{N}$, $k < m$, $\alpha, \beta \in [0, 1]$. Let $\mathcal{X}$ be a set, let $A: (\mathcal{X} \times \{\pm 1\})^* \to \{\pm 1\}^{\mathcal{X}}$ be a learning rule, and let $\mathbb{D} \subseteq \Delta(\mathcal{X} \times \{\pm 1\})$. We say that $\underline{A \text{ is } (\alpha, \beta, m, k)\text{-hypothesis}}$ $\underline{\text{stable with respect to } \mathbb{D}}$ if*

$$\forall \mathcal{D} \in \mathbb{D}: \mathop{\mathbb{P}}_{\substack{S_1 \sim \mathcal{D}^{m-k} \\ S_2 \sim \mathcal{D}^k}} [\mathrm{dist}_{\mathcal{D}_{\mathcal{X}}}(A(S_1), A(S_1 \circ S_2)) \leq \alpha] \geq 1 - \beta,$$

*where $\mathcal{D}_{\mathcal{X}}$ is the marginal of $\mathcal{D}$ on $\mathcal{X}$, $\mathrm{dist}_{\mathcal{P}}(f, g) = \mathbb{P}_{x \sim \mathcal{P}}[f(x) \neq g(x)]$, and $\circ$ denotes concatenation.*

**Definition 4.2.** *In the notation of Definition 4.1, we say that $\underline{A \text{ is } (\alpha, \beta, m, k)\text{-loss stable}}$ $\underline{\text{with respect to } \mathbb{D}} \text{ if } \forall \mathcal{D} \in \mathbb{D}: \mathbb{P}_{S_1 \sim \mathcal{D}^{m-k}, S_2 \sim \mathcal{D}^k} \left[ \left| L_{\mathcal{D}}(A(S_1)) - L_{\mathcal{D}}(A(S_1 \circ S_2)) \right| \leq \alpha \right] \geq 1 - \beta.$*

**Theorem 4.3.** *Let $k \in \mathbb{N}$ and $\alpha_0, \beta_0 \in (0, 1)$ such that $k \geq \Omega\big(\log(1/\beta_0)/\alpha_0^2\big)$. Let $A$ be a learning rule that is $(\alpha_1, \beta_1, m, k)$-hypothesis stable or loss stable with respect to $\mathbb{D}$ (as in Definitions 4.1 and 4.2). Then $(A, \mathbb{D})$ is $(\varepsilon = \alpha_0 + \alpha_1, \delta = \beta_0 + \beta_1, m)$-uniformly estimable.*

The proof of Theorem 4.3 appears in Appendix G.

Hence, stability is a sufficient condition for estimability. We remark that it is not a necessary condition. For instance, a learning rule selected at random as in Example 1.8 most likely is

---

[12]Indeed, if a setting is not overparameterized, then estimability is typically easy due to standard uniform convergence bounds for VC classes (specifically, for proper learning rules in realizable settings; see Example 1.5 for a counterexample when the algorithm is not proper). So when discussing inestimability, we focus on overparameterized settings. See further discussion in Appendix B.

[13]In both cases, the parameters we state are not tight, and the settings are actually more overparameterized than stated.

[14]The memorization algorithm always outputs the function $h(x) = -1$, except that it alters $h$ in a small number of locations to fit the training set.

estimable (because it has high loss for any distribution), but not hypothesis stable (since for each possible input sample, it outputs a different hypothesis that was chosen at random). To see that loss stability is also not necessary for estimability, fix a degenerate distribution $\mathcal{D}$ such that $\mathcal{D}((x^*, 1)) = 1$ for some $x^*$, and consider an algorithm $A$ that for samples of size $m$ outputs the constant hypothesis $h_1(x) = 1$, and for samples of size $m - k$ outputs the constant hypothesis $h_{-1}(x) = -1$. $A$ is perfectly estimable with respect to $\{\mathcal{D}\}$, but it is not loss stable.

One might object that Theorem 4.3 is of limited utility, because it is hard to check whether a given algorithm is hypothesis stable or loss stable. Our response to this criticism is that in practice, it is quite easy to check whether an algorithm is loss (or hypothesis) stable with respect to a particular population distribution – and indeed we do so in our experiments (see Appendix M).

The process for estimating loss stability is simple: take a set $S$ of $m$ i.i.d. labeled samples from the population distribution. Randomly choose a subset $S'$ of size $m - k$. Execute the learning algorithm twice, once with training set $S$ to produce a hypothesis $h$, and another time with training set $S'$ to produce a hypothesis $h'$. Use an additional validation set to estimate the difference in population loss between $h$ and $h'$. Repeating this process a number of times and taking an average gives a good estimate of the $(m, k)$-loss stability. A similar process can be used to estimate hypothesis stability. Simply measure the disagreement between $h$ and $h'$ on the validation set (note that in this case, the validation set can be unlabeled, which is an advantage when labeling data is expensive).

## 5 A Simple Characterization

The following definition is a variant of Definition 1.2. Such a variant allows us to have a simple characterization of estimability in Fact 5.2. Namely, to understand whether an algorithm is estimable with respect to a set of distributions, one can examine the quantity $\mathbb{E}_{\mathcal{D} \sim \mathrm{U}(\mathbb{D}), S \sim \mathcal{D}^m}[\mathrm{var}(L_{\mathcal{D}}(A(S)) \mid S)]$.

**Definition 5.1.** *Let $\mathbb{D}$ be a set of distributions and let $A$ be a learning algorithm. We say that $A$ is $(\varepsilon, m)$-estimable in $\ell_2$ with respect to $\mathbb{D}$, if there exists an estimator $\mathcal{E}$ such that*

$$\mathbb{E}_{\mathcal{D} \sim \mathrm{U}(\mathbb{D}), S \sim \mathcal{D}^m} \left[ (\mathcal{E}(S) - L_{\mathcal{D}}(A(S)))^2 \right] \leq \varepsilon$$

We remark that for bounded loss functions, one can move between Definition 5.1 and Definition 1.2 using Markov's inequality. Furthermore, although the characterization in the following theorem is simple, it might provide a technical condition that will be useful for future work.

**Fact 5.2.** *$A$ is $(\varepsilon, m)$-estimable in $\ell_2$ with respect to $\mathbb{D}$ if and only if*

$$\mathbb{E}_{\mathcal{D} \sim \mathrm{U}(\mathbb{D}), S \sim \mathcal{D}^m}[\mathrm{var}(L_{\mathcal{D}}(A(S)) \mid S)] \leq \varepsilon.$$

The proof of Fact 5.2 appears in Appendix H.

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

## A  OTHER RELATED WORKS

The works of Nagarajan and Kolter (2019, Theorem 3.1) and Bartlett and Long (2021, Theorem 1) also study cases where generalization bounds fall short of estimating the performance of learning algorithms (while Negrea et al., 2020 provide a response to these claims). They preclude tight algorithm-dependent generalization bounds only for uniform convergence and linear classifiers. Their theorems consider specific distributions (Gaussian in Nagarajan and Kolter, 2019, a different distribution per sample in Bartlett and Long,

2021) and specific types of SGD. In contrast, our work uses the same marginal distribution across all sample sizes, and applies to many algorithms and distributions.

We now mention a few of the algorithm-dependent generalization bounds in the literature. Zhang, Teng, and Zhang (2023) study convex optimization, so their results apply only to a single neuron. While providing matching lower and upper bounds, these bounds match only asymptotically when the sample size $n$ is very large, far from the overparameterized regime relevant for neural networks. Nikolakakis, Haddadpour, Karbasi, and Kalogerias (2023) proposes generalization bounds for algorithms satisfying a certain symmetry property (e.g., full-batch gradient descent) when using smooth losses. These bounds are algorithm-dependent but distribution-free, making no distributional assumptions.

There are a number of information-theoretic generalization bounds that are both algorithm and distribution-dependent, such as Theorem 1 of Xu and Raginsky, 2017. However, such bounds are sometimes difficult to approximate numerically in a tight manner. These bounds are part of the PAC-Bayes framework.[15] Unfortunately, when these PAC-Bayes or information-theoretic bounds can be approximated in a tight manner,[16] they do not reveal what properties of the (distribution, algorithm) pair allowed for such success in learning and estimation. The works of Haghifam, Moran, Roy, and Dziugiate (2022b) and Rammal, Achille, Golatkar, Diggavi, and Soatto (2022) use the notion of leave-one-out conditional mutual information to derive generalization bounds, which provide another characterization of VC classes and yield non-vacuous generalization bounds for neural networks.

### A.1 Neural Tangent Kernel and Mean-Field Theory

There are many works that study generalization in the overparameterized regime using the neural tangent kernel (NTK) or mean-field theory (MFT) approach.[17] To the best of our knowledge, these works do not provide general necessary or sufficient conditions for generalization bounds to be tight, which is the focus of our work. Additionally, they study generalization bounds for fairly specific families of algorithms such as gradient descent (or idealized versions thereof), while our work applies to a broader and more general class of algorithms.[18]

## B On the Definition of Overparameterization

In this paper we use a definition of overparameterization (Definition 1.4) introduced by Gastpar et al. (2024) (see Definition 2 in that paper).

This definition is motivated by the need to formalize many common definitions of overparameterization in a manner that is true to the basic intuitive notion of overparameterization while also being is suitable for proving mathematical theorems.

Appendix D[19] of Gastpar et al. (2024) offers a detailed discussion of the merits of Definition 1.4, and how it generalizes many common definitions. Below, we provide a brief summary of the main points made in that appendix (the reader is encouraged to consult the original appendix for a more detailed discussion).

Gastpar et al. (2024) identify three informal notions of overparameterization that are common in the literature:

---

[15]See proof 4 for Theorem 8 in Bassily et al. (2018), which is equivalent to Theorem 1 in Xu and Raginsky (2017).

[16]Such as in Issa et al. (2019), Issa et al. (2023), Esposito et al. (2021), Harutyunyan et al. (2021), Dziugaite et al. (2021), Haghifam et al. (2022a), Hellström and Durisi (2022), and Wang and Mao (2023).

[17]E.g., Aminian et al. (2023), Chen et al. (2020), Nishikawa et al. (2022), and Nitanda et al. (2021).

[18]We note that Theorems 3.1 and 3.2 apply to learning algorithms that achieve 0 training error. Because this property is satisfied by most contemporary overparameterized learning algorithms (even if the labels are random), we do not view this as a significant limitation on the generality of our results. This assumption is not essential, and it could easily be relaxed in future work.

[19]This is Appendix D of the official ICLR version of the paper, linked to in our References section. The numbering of appendices might differ in other versions.

**Definition A.** The number of independently-tunable parameters in the machine learning system is significantly larger than the number of samples in the training set.

**Definition B.** The learning system can interpolate arbitrary data.

**Definition C.** The size of the training set is smaller than the VC dimension.

The authors argue that each of these definitions, as typically understood, implies their proposed formalization of overparameterization as in Definition 1.4. In particular, this means that any impossibility result proved for overparameterized settings as in Definition 1.4 also hold for settings that are overparameterized according to any of the above three definitions.

We are specifically interested in understanding estimability in the overparameterized setting, both because most contemporary machine learning systems are overparameterized, and also because estimability in the standard (non-overparameterized) supervised learning setting is typically trivial due to standard uniform convergence bounds for VC classes (specifically, for proper learning rules in realizable settings; see Example 1.5 for a counterexample when the algorithm is not proper).

Overparameterization as in Definition 1.4 appears in the current paper in the following contexts:

1. In Theorems 3.1 and 3.2, show that show cases where algorithms are not estimable. These results are in the overparameterized setting.

2. In Theorem 4.3 we give a condition that implies that an algorithm is estimable, even if the setting is overparameterized.

3. In Fact 5.2 we give a necessary and sufficient condition for estimability, including in the overparameterized setting.

**Remark B.1.** *One way to understand the motivation for Definition 1.4 is to consider an analogy with the definition of a continuous function. The most common and basic definition of a continuous function is the $\varepsilon$-$\delta$ definition (developed by Bolzano, Cauchy, Weierstrass and Jordan in the 1800s). That definition very clearly captures the intuitive notion of continuity. Later on, however, that definition was generalized by Hausdorff, who gave a modern topological definition of a continuous function, requiring that the preimage of any open set is an open set. This more modern definition might appear rather strange at first, and somewhat removed from the basic intuition of what a continuous function is. Nonetheless, it turns out that the topological definition does not just generalize the basic definition, but it does so in a way that is very useful, while remaining true to the intuitive notion of continuity. Similarly, Definition 1.4 generalizes the intuitive notion of overparameterization in a way that is useful for proving mathematical theorems.*

## C  ON EXTENDING OUR RESULTS TO RANDOMIZED ALGORITHMS

For simplicity, in this paper we focus on deterministic learning rules. However, we recognize that the topic of randomized learning algorithms is very important, seeing as most algorithms used in practice today are randomized.

The estimability framework explored in this paper can be extended to handle randomized algorithms as well, and in fact the original work of Gastpar et al. (2024) already contains some initial treatment of randomized algorithms.

We expect that the results presented in this paper can be extended to randomized algorithms, and that the essence of the results remains mostly unchanged.

The first step in such an extension would be to clearly define what estimability means for randomized learning algorithms. A definition that one might initially consider is one where the estimator knows the randomness used by the algorithm, and must output a number that is with high probability close to the true population loss of the randomized algorithm. This definition is not very interesting, because a setting in which the estimator knows the randomness used by the randomized algorithm is equivalent to the setting of a deterministic algorithm, which is already covered by the results in this paper. Nonetheless, it is good to keep this definition in mind, because it means that our results for deterministic algorithms

already apply as-is to randomized algorithms (like SGD) once the randomly chosen seed is fixed, which might be a simple and satisfactory approach for many purposes (SGD with a fixed random seed typically performs as well for most purposes as SGD with a fresh randomly-chosen seed).

Perhaps the more "correct" and interesting definition of estimability for randomized learning algorithms is one where the estimator knows the training set, but does not know the randomness used by the learning algorithm, and it is required to output a number that is close with high probability to the *expected* population loss of the randomized algorithm when executed with this training set (where the expectation is over the randomness of the algorithm). In this setting, we believe the essence of our results carries through, with an important conceptual difference: using randomness, one can always engineer a learning algorithm that is estimable, essentially by adding noise to the output of the algorithm. As the noise in the algorithm's output increases, the expected 0-1 loss of the algorithm becomes closer to $1/2$, and so the algorithm becomes estimable with a trivial estimator that simply always outputs the number $1/2$. (With intermediate amounts of noise, a number between 0 and $1/2$ will be optimal).

Consequently, for randomized algorithms, our lower bounds in Theorems 3.1 and 3.2 can no longer be stated as absolute limitations on estimability. Rather there is now a trade-off between the performance of the algorithm and its estimability. As one adds more noise, the algorithm becomes more estimable, but its performance degrades. Thus, the corresponding theorems for randomized algorithm would state that no algorithm can simultaneously make good predictions for some large set of labeling functions and also be estimable.

On the other hand, the upper bound in Theorem 4.3 that states that stable algorithms are estimable remains basically unchanged for randomized algorithms.

To summarize, under a suitable definition of estimability for randomized algorithms, we expect that our results would not change much, though the statement (and proof) of the lower bounds would be somewhat more complex. We leave this work to future research.

## D    DETAILS ABOUT EXAMPLE 1.9

For sample size $m \geq d + 10$, any ERM algorithm for $\mathcal{H}$ satisfies $p(m) \geq 0.999$, meaning it learns $\mathbb{D}$ well, and hence is $(0, 10^{-3}, d + 10)$-estimable on average. This holds because for an ERM to output the ground truth, it is clearly sufficient that only a single sample-consistent function exists in the concept class (the ground truth). Similarly, in the event that there are $t > 1$ sample-consistent functions, the success probability is given by $1/t$ due to the uniform prior over ground truth distributions. Parity functions are fully characterized by their coefficient vector $w = [w_1, \dots, w_d]$. Since the labels $y$ are a bilinear function in the inputs $x$ and coefficients $w$, one can obtain $w$ from $m \geq d$ linearly independent samples $x_i$ by solving the linear system of equation $y = Xw$ with design matrix $X \in \{0, 1\}^{m \times d}$. More generally, $X$ having rank $d - k$ is equivalent to the event of having $t = 2^k$ sample-consistent functions (coefficient vectors) since every additional linearly independent row rules out half of all parity functions. Now assume $X$ consists of all i.i.d. Ber($1/2$) entries and $y$ contains the labels of all samples. The probability of zero population loss can now be obtained from the law of total probability with the probabilities of rank deficiency computed according to Corollary 2.2 in Blake and Studholme (2006).

Similar calculations show that for smaller sample sizes, any ERM for $\mathcal{H}$ satisfies $p(d) \geq 0.61$, and $p(d - 1) \geq 0.38$. An application of Theorem 5 in Gastpar et al. (2024) shows that there exist ERM algorithms such that for any $6 \leq m \leq d$ there exists a collection $\mathbb{D}_m$ for which the algorithm is not $(0.25, 0.32, m)$-estimable on average. These algorithms have an inductive bias towards a subset $\mathcal{F} \subseteq \mathcal{H}$, such that they perform well for distributions labeled by a function from $\mathcal{F}$, and perform poorly for target functions from the complement of $\mathcal{F}$.

# E    Proof of Theorem 3.1

Recall the definition of nearly-orthogonal functions (Definition 2.8). The proof of Theorem 3.1 uses a corollary of the Johnson–Lindenstrauss lemma (Theorem L.1), which states that random vectors in a high dimensional space are nearly orthogonal, as follows.[20]

**Claim E.1.** *Let $\varepsilon \in (0, 1/2)$, and let $d, n \in \mathbb{N}$ such that*

$$n \le \exp\big(d\varepsilon^2 / 54\big).$$

*Let $\mathcal{U} = \mathrm{U}\big(\{\pm 1\}^{[d]}\big)$ be the uniform distribution over functions $[d] \to \{\pm 1\}$, and consider a random sequence $F$ of functions $F_1, \ldots, F_n$ sampled independently from $\mathcal{U}$. Then*

$$\mathbb{P}_{F \sim \mathcal{U}^n}\big[F \in \perp_{\varepsilon, [d]}\big] \ge 0.99.$$

*Proof of Claim E.1.* If $n = 1$ there is nothing to prove, so we assume $n \ge 2$. Let $R \sim \mathrm{U}\big(\{\pm 1\}^{d \times n}\big)$ be a $d \times n$ matrix with entries in $\{\pm 1\}$ chosen independently and uniformly at random. In particular, for each $i \in [n]$, the $i$-th column of $R$ is a vector of $d$ numbers in $\{\pm 1\}$ chosen independently and uniformly at random. Hence, using $e_1, \ldots, e_n$ to denote the standard basis of $\mathbb{R}^n$, we identify the vector $Re_i$, which is the $i$-th column of $R$, with the random function $F_i : [d] \to \{\pm 1\}$.

Recall that for vectors $u, v \in \mathbb{R}^d$,

$$\|u - v\|_2^2 = \langle u - v, u - v \rangle = \|u\|_2^2 - 2\langle u, v \rangle + \|v\|_2^2,$$

so

$$\langle u, v \rangle = \frac{\|u\|_2^2 + \|v\|_2^2 - \|u - v\|_2^2}{2}. \tag{5}$$

Invoking Theorem L.1 with $s = n$, $\beta = 7$, $V = \{e_1, \ldots, e_n\} \subseteq \mathbb{R}^n$, and $d, n, \varepsilon$ as in the claim statement implies that

$$\mathbb{P}_{R \sim \mathrm{U}(\{\pm 1\}^{d \times n})} \left[ \begin{array}{l} \forall i, j \in [n], i \ne j : \\[4pt] (1 - \varepsilon) \cdot 2 \le \left\| \frac{1}{\sqrt{d}} Re_i - \frac{1}{\sqrt{d}} Re_j \right\|_2^2 \le (1 + \varepsilon) \cdot 2 \end{array} \right] \ge 1 - \frac{1}{n^\beta}. \tag{6}$$

Hence, with probability at least $1 - 1/n^\beta \ge 1 - 1/2^7 \ge 0.99$ over the choice of $F$, every distinct $i, j \in [n]$ satisfy

$$\left| \mathbb{E}_{x \sim \mathrm{U}([d])}[F_i(x)F_j(x)] \right| = \left| \frac{1}{d} \sum_{x \in [d]} F_i(x)F_j(x) \right| = \left| \frac{1}{d} \langle Re_i, Re_j \rangle \right| \quad \text{(Identifying } F_i \text{ with } Re_i)$$

$$= \left| \frac{\|Re_i\|_2^2 + \|Re_j\|_2^2 - \|Re_i - Re_i\|_2^2}{2d} \right| \quad \text{(By Eq. (5))}$$

$$= \left| 1 - \frac{1}{2} \left\| \frac{1}{\sqrt{d}} Re_i - \frac{1}{\sqrt{d}} Re_i \right\|_2^2 \right|$$

$$\le \varepsilon, \quad \text{(By Eq. (6))}$$

as desired. $\qquad\qquad\qquad\qquad\qquad\qquad\qquad\qquad\qquad\qquad\qquad\qquad\qquad\qquad\qquad\square$

*Proof of Theorem 3.1.* Fix an $\mathcal{H}$-shattered set $\mathcal{X}_d \subseteq \mathcal{X}$ with cardinality $|\mathcal{X}_d| = d$, and for each $f : \mathcal{X}_d \to \{\pm 1\}$ let $\mathcal{D}_f = \mathrm{U}(\{(x, f(x)) : x \in \mathcal{X}_d\})$. Note that the distributions $\mathcal{D}_f$ are $\mathcal{H}$-realizable. We will show that there exists a collection $\mathbb{D} = \{\mathcal{D}_f : f \in \mathcal{F}\}$ that satisfies Eq. (4), where $\mathcal{F} \subseteq \{\pm 1\}^{\mathcal{X}_d}$ is a set of $k = 2^m + 1$ functions.

Consider the following experiment:

---

[20]It is also possible to prove a similar claim by directly using concentration of measure (e.g., Hoeffding's inequality), without using the Johnson–Lindenstrauss lemma.

1. Sample a sequence of functions $G = (G_1, \ldots, G_k)$ independently and uniformly at random from $\{\pm 1\}^{\mathcal{X}_d}$.

2. Sample a function $F$ uniformly from $G$.

3. Sample a sequence of points $X = (X_1, \ldots, X_m)$ independently and uniformly at random from $\mathcal{X}_d$. ($X$ is sampled independently of $(G, F)$.)

4. For each $i \in [m]$, let $Y_i = F(X_i)$, let $Y = (Y_1, \ldots, Y_m)$, and let $S = \Big( (X_1, Y_1), \ldots, (X_m, Y_m) \Big)$.

Let $\mathcal{P}$ be the joint distribution of $(G, F, X, Y, S)$. Consider the following events:

- $\mathcal{E}_1 = \{G \in \perp_{\varepsilon, \mathcal{X}_d}\}$ for $\varepsilon = 2/d^{1/4}$. By Claim E.1 and the choice of $k$, $\mathcal{P}(\mathcal{E}_1) \geq 0.99$ for $d$ large enough.[21]

- $\mathcal{E}_2 = \big\{ |\{X_1, \ldots, X_m\}| = m \big\}$. By Claim L.2 and the choice of $m$, $\mathcal{P}(\mathcal{E}_2) \geq 0.99$.

- $\mathcal{E}_3 = \{|G_S| = 2\}$. $\mathcal{P}(\mathcal{E}_3 \mid \mathcal{E}_2) \geq 1/e$. To see this, note that each function $G_i \in G \setminus \{F\}$ is chosen independently of $F$. Hence, the probability that a function $G_i$ agrees with $F$ on the $m$ distinct samples in $X$ (i.e., the probability that $G_i(X_j) = F(X_j)$ for all $j \in [m]$, given $\mathcal{E}_2$) is $p = 2^{-m}$. The functions in $G$ are chosen independently, so the number $T$ of functions in $G \setminus \{F\}$ that agree with $F$ on $m$ distinct samples has a binomial distribution $T \sim \mathrm{Bin}(k-1, p)$. So

$$\mathbb{P}[T = 1] = (k-1) \cdot p \cdot (1-p)^{k-2} = (1-p)^{k-2}$$
$$\geq \left( e^{-\frac{p}{1-p}} \right)^{k-2} \qquad\qquad (\forall p < 1 : \ 1 - p \geq e^{-p/(1-p)})$$
$$= 1/e.$$

Let $\mathcal{E} = \mathcal{E}_1 \cap \mathcal{E}_3$. Combining the above bounds yields

$$\mathcal{P}(\mathcal{E}) = \mathcal{P}(\mathcal{E}_1 \cap \mathcal{E}_3)$$
$$\geq \mathcal{P}(\mathcal{E}_3) - \mathcal{P}\big(\mathcal{E}_1^C\big)$$
$$\geq \mathcal{P}(\mathcal{E}_3 \mid \mathcal{E}_2) \cdot \mathcal{P}(\mathcal{E}_2) - \mathcal{P}\big(\mathcal{E}_1^C\big)$$
$$\geq 0.99 \cdot 1/e - 0.01 > 1/3.$$

By an averaging argument, this implies that there exists $\mathcal{F} \subseteq \{\pm 1\}^{\mathcal{X}_d}$ such that $\mathcal{F} \in \perp_{\varepsilon, \mathcal{X}_d}$ for $\varepsilon = 2/d^{1/4}$ and

$$\mathcal{P}(|G_S| = 2 \mid G = \mathcal{F}) \geq 1/3. \tag{7}$$

Fix this $\mathcal{F}$, and let $A$ be an $\mathcal{F}$-interpolating learning rule. From the technical lemma (Lemma I.1), there exists a collection of $\mathcal{F}$-realizable distributions $\mathbb{D} \subseteq \Delta(\mathcal{X}_d \times \{\pm 1\})$ such that for any estimator $\mathcal{E} : (\mathcal{X} \times \{\pm 1\})^m \to [0, 1]$ that may depend on $\mathbb{D}$ and $A$,

$$\mathbb{P}_{\substack{\mathcal{D} \sim \mathrm{U}(\mathbb{D}) \\ S \sim \mathcal{D}^m}}\left[ |\mathcal{E}(S) - L_\mathcal{D}(A(S))| \geq \frac{1}{4} - \frac{\varepsilon}{4} \right] \geq \frac{1}{2} \cdot \mathbb{P}_{\substack{\mathcal{D} \sim \mathrm{U}(\mathbb{D}) \\ S \sim \mathcal{D}^m}}[|\mathcal{F}_S| = 2]$$
$$\geq \frac{1}{2} \cdot \frac{1}{3} = \frac{1}{6}, \qquad\qquad \text{(By Eq. (7))}$$

as desired. $\qquad\qquad\qquad\qquad\qquad\qquad\qquad\qquad\qquad\qquad\qquad\qquad\qquad\qquad\qquad\square$

## F  PROOF OF THEOREM 3.2

*Proof of Theorem 3.2.* We take $\mathbb{D} = \{\mathcal{D}_f : f \in \mathcal{F}\}$ where $\mathcal{D}_f = \mathrm{U}(\{(x, f(x)) : x \in \mathcal{X}\})$. Fix a function $f^* \in \mathcal{F}$, let $S \sim (\mathcal{D}_{f^*})^m$, and consider the random variable $Z = |\mathcal{F}_S|$. We bound the expectation and variance of $Z$, and then show a lower bound on the probability that $Z \in \{2, 3\}$.

---

[21]We choose $d_0 \in \mathbb{N}$ to be the universal constant such that this inequality holds for all integers $d \geq d_0$ and all $m \leq \sqrt{d}/10$.

Let $S = \big((X_1, Y_1), \ldots, (X_m, Y_m)\big)$ and $X = \{X_1, \ldots, X_m\}$, and let $E$ denote the event in which $|X| = m$ (i.e., $S$ is collision-free). For each $f \in \mathcal{F}$, let $Z_f = \mathbb{1}(\forall i \in [m]: f(X_i) = Y_i)$, so that $Z = \sum_{f \in \mathcal{F}} Z_f$.

$$
\mathop{\mathbb{E}}_{S \sim (\mathcal{D}_{f^*})^m}[Z \mid E] = \mathbb{E}\left[\sum_{f \in \mathcal{F}} Z_f \;\middle|\; E\right]
$$

$$
= 1 + \sum_{\substack{f \in \mathcal{F} \\ f \neq f^*}} \mathbb{P}\big[\forall i \in [m]: f(X_i) = Y_i \mid E\big] \qquad (Z_F = 1)
$$

$$
\leq 1 + 2^m \cdot \left(\frac{1}{2} + \frac{1}{2} \cdot \frac{1}{1000m}\right)^m \qquad (\text{By Fact 2.9})
$$

$$
\leq 1 + e^{1/1000} < 2.002. \qquad (8)
$$

$$
\mathop{\mathbb{E}}_{S \sim (\mathcal{D}_{f^*})^m}[Z \mid E] \geq 1 + 2^m \cdot \left(\frac{1}{2} - \frac{1}{2} \cdot \frac{1}{1000m}\right)^m \qquad (\text{By Fact 2.9})
$$

$$
\geq 1 + e^{-1/500}. \qquad (1 - x \geq e^{-x/(1-x)})
$$
$$
(9)
$$

$$
\mathop{\mathbb{E}}_{S \sim (\mathcal{D}_{f^*})^m}\big[Z^2 \mid E\big] = \mathbb{E}\left[\left(\sum_{f \in \mathcal{F}} Z_f\right)\left(\sum_{g \in \mathcal{F}} Z_g\right) \;\middle|\; E\right]
$$

$$
= \mathbb{E}\left[\left(1 + \sum_{\substack{f \in \mathcal{F} \\ f \neq f^*}} Z_f\right)\left(1 + \sum_{\substack{g \in \mathcal{F} \\ g \neq f^*}} Z_g\right) \;\middle|\; E\right] \qquad (Z_{f^*} = 1)
$$

$$
= \mathbb{E}\left[1 + 2\sum_{\substack{f \in \mathcal{F} \\ f \neq f^*}} Z_f + \sum_{\substack{f \in \mathcal{F} \\ f \neq f^*}}\sum_{\substack{g \in \mathcal{F} \\ g \neq f^*}} Z_f Z_g \;\middle|\; E\right]
$$

$$
= \mathbb{E}\left[1 + 3\sum_{\substack{f \in \mathcal{F} \\ f \neq f^*}} Z_f + \sum_{\substack{f, g \in \mathcal{F} \setminus \{f^*\} \\ f \neq g}} Z_f Z_g \;\middle|\; E\right]
$$

$$
= 1 + 3\,(\mathbb{E}[Z \mid E] - 1) + \sum_{\substack{f, g \in \mathcal{F} \setminus \{f^*\} \\ f \neq g}} \mathbb{E}\Big[Z_f Z_g \;\Big|\; E\Big]. \qquad (10)
$$

$$
\sum_{\substack{f, g \in \mathcal{F} \setminus \{f^*\} \\ f \neq g}} \mathbb{E}\Big[Z_f Z_g \;\Big|\; E\Big] = \sum_{\substack{f, g \in \mathcal{F} \setminus \{f^*\} \\ f \neq g}} \mathbb{P}\big[\forall i \in [m]: f(X_i) = g(X_i) = f^*(X_i) \mid E\big]
$$

$$
\leq 2^{2m} \cdot \left(\frac{1}{4} + \frac{3}{4} \cdot \frac{1}{1000m}\right)^m \qquad (\text{By Claim K.1})
$$

$$
= \left(1 + \frac{3}{1000m}\right)^m
$$

$$
\leq e^{3/1000}. \qquad (11)
$$

Combining Eqs. (8) to (11) yields

$$
\begin{aligned}
\mathrm{Var}[Z \mid E] &= \mathbb{E}\big[Z^2 \mid E\big] - \big(\mathbb{E}[Z \mid E]\big)^2 \\
&\leq 1 + 3e^{1/1000} + e^{3/1000} - \left(1 + e^{-1/500}\right)^2 \\
&< 1.02.
\end{aligned}
$$

By Lemma J.1,

$$
\mathbb{P}[Z \in \{2,3\} \mid E] \geq 1 - \frac{\mathrm{Var}[Z \mid E]}{2} \geq 0.49.
$$

Claim L.2 and $|\mathcal{X}'| \geq 100m^2$ imply that $\mathbb{P}[E] \geq 0.99$. Hence,

$$
\mathbb{P}[Z \in \{2,3\}] \geq \mathbb{P}[E] \cdot \mathbb{P}[Z \in \{2,3\} \mid E] \geq 0.99 \cdot 0.49 \geq 0.48. \tag{12}
$$

Finally, invoking our technical lemma (Lemma I.1) yields

$$
\mathbb{P}_{\substack{F \sim \mathrm{U}(\mathcal{F}) \\ S \sim (\mathcal{D}_F)^m}}\left[\big|\mathcal{E}(S) - L_{\mathcal{D}_F}(A(S))\big| \geq \frac{1}{4} - \frac{1}{4000m}\right] \geq \frac{\mathbb{P}[Z \in \{2,3\}]}{3} \geq 0.16,
$$

as desired. $\qquad\square$

## G   PROOF OF THEOREM 4.3

*Proof of Theorem 4.3.* If $(A, \mathbb{D})$ is $(\alpha, \beta, m, k)$-hypothesis stable, then in particular $(A, \mathbb{D})$ is also $(\alpha, \beta, m, k)$-loss stable. Hence, it suffices to prove the claim for the case of loss stability. We construct a uniform estimator $\mathcal{E}$ as follows. Given a sample $S \in \mathcal{Z}^m$ for $\mathcal{Z} = (\mathcal{X} \times \{\pm 1\})$, let $S_1 \circ S_2 = S$ be the partition of $S$ such that $S_1 \in \mathcal{Z}^{m-k}$ and $S_2 \in \mathcal{Z}^k$. Take $\mathcal{E}(S) = L_{S_2}(A(S_1))$.

By the triangle inequality,

$$
|\mathcal{E}(S) - L_{\mathcal{D}}(A(S))| \leq |\mathcal{E}(S) - L_{\mathcal{D}}(A(S_1))| + |L_{\mathcal{D}}(A(S_1)) - L_{\mathcal{D}}(A(S))|,
$$

so

$$
\begin{aligned}
\mathbb{P}_{S \sim \mathcal{D}^m}[|\mathcal{E}(S) - L_{\mathcal{D}}(A(S))| > \varepsilon] &\leq \mathbb{P}\left[\begin{array}{l}|L_{S_2}(A(S_1)) - L_{\mathcal{D}}(A(S_1))| > \alpha_0 \vee \\ |L_{\mathcal{D}}(A(S_1)) - L_{\mathcal{D}}(A(S))| > \alpha_1\end{array}\right] \\
&\leq \mathbb{P}[|L_{S_2}(A(S_1)) - L_{\mathcal{D}}(A(S_1))| > \alpha_0] \\
&\qquad + \mathbb{P}[|L_{\mathcal{D}}(A(S_1)) - L_{\mathcal{D}}(A(S))| > \alpha_1] \\
&\leq \beta_0 + \beta_1 = \delta,
\end{aligned}
$$

where the final step follows from Hoeffding's inequality, the choice of $k$, and the stability of $A$. $\qquad\square$

## H   PROOF OF FACT 5.2

*Proof.* The result that the minimum mean-square error (MMSE) estimator corresponds to the conditional expectation is a well-established theorem in probability theory (see, for instance, Section 7.9 in Grimmett and Stirzaker (2020)). For the sake of completeness, we present a proof of this result.

We will use the following simple claim.

**Claim H.1.** *Let $c_1, ..., c_k, p_1, ..., p_k \in \mathbb{R}$ such that $\sum_{i=1}^k p_i = 1$, then*

$$
\mathrm{argmin}_{x \in \mathbb{R}} \sum_{i=1}^k p_i \cdot (x - c_i)^2 = \sum_{i=1}^k p_i \cdot c_i.
$$

The claim follows by taking the derivative of $\sum_{i=1}^{k} p_i \cdot (x - c_i)^2$ with respect to $x$ which yields the equation:

$\sum_{i=1}^{k} 2p_i(x - c_i) = 0$ that implies $x = \sum_{i=1}^{k} p_i c_i$ since $\sum_{i=1}^{k} p_i = 1$.

The following shows that the estimator $\mathcal{E}^*(S) := \mathbb{E}\left[L_{\mathcal{D}}(A(S)) \mid S\right]$ is optimal and the inequality follows from Claim H.1. Let $\mathcal{E}$ be any estimator for $A$.

$$
\mathbb{E}_{\mathcal{D}\sim U(\mathbb{D}),S\sim\mathcal{D}^m}\left[(\mathcal{E}(S) - L_{\mathcal{D}}(A(S)))^2\right]
$$

$$
= \sum_S \mathbb{P}(S) \sum_{\mathcal{D}\in\mathbb{D}} \mathbb{P}(\mathcal{D}|S)\left(L_{\mathcal{D}}(A(S)) - \mathcal{E}(S)\right)^2
$$

$$
= \mathbb{E}\left[\sum_{\mathcal{D}\in\mathbb{D}} \mathbb{P}(\mathcal{D}|S)\left(L_{\mathcal{D}}(A(S)) - \mathcal{E}(S)\right)^2\right]
$$

$$
\geq \mathbb{E}\left[\sum_{\mathcal{D}\in\mathbb{D}} \mathbb{P}(\mathcal{D}|S)\left(L_{\mathcal{D}}(A(S)) - \sum_{\mathcal{D}\in\mathbb{D}}[\mathbb{P}(\mathcal{D}|S)L_{\mathcal{D}}(A(S))]\right)^2\right]
$$

$$
= \mathbb{E}\left[\sum_{\mathcal{D}\in\mathbb{D}} \mathbb{P}(\mathcal{D}|S)\left(L_{\mathcal{D}}(A(S)) - \mathcal{E}^*(S)\right)^2\right]
$$

$$
= \mathbb{E}_{\mathcal{D}\sim U(\mathbb{D}),S\sim\mathcal{D}^m}\left[(\mathcal{E}^*(S) - L_{\mathcal{D}}(A(S)))^2\right].
$$

This means that $A$ is square loss $(\varepsilon, m)$-estimable with respect to $\mathbb{D}$ if and only if $\mathcal{E}^*$ can achieve $\varepsilon$ accuracy. It achieves such accuracy if and only if $\mathbb{E}\left[\text{var}(L_{\mathcal{D}}(A(S)) \mid S)\right] \leq \varepsilon$. This follows by the following equalities that complete the proof.

$$
\mathbb{E}\left[\text{var}(L_{\mathcal{D}}(A(S)) \mid S)\right] = \mathbb{E}\left[\mathbb{E}\left[\left(L_{\mathcal{D}}(A(S)) - \mathbb{E}\left[L_{\mathcal{D}}(A(S))|S\right]\right)^2 \mid S\right]\right]
$$

$$
= \mathbb{E}\left[\sum_{\mathcal{D}\in\mathbb{D}} \mathbb{P}(\mathcal{D}|S)\left(L_{\mathcal{D}}(A(S)) - \mathbb{E}\left[L_{\mathcal{D}}(A(S)) \mid S\right]\right)^2\right]
$$

$$
= \mathbb{E}\left[\sum_{\mathcal{D}\in\mathbb{D}} \mathbb{P}(\mathcal{D}|S)\left(L_{\mathcal{D}}(A(S)) - \sum_{\mathcal{D}\in\mathbb{D}}[\mathbb{P}(\mathcal{D}|S)L_{\mathcal{D}}(A(S))]\right)^2\right]
$$

$$
= \mathbb{E}\left[\sum_{\mathcal{D}\in\mathbb{D}} \mathbb{P}(\mathcal{D}|S)\left(L_{\mathcal{D}}(A(S)) - \mathcal{E}^*(S)\right)^2\right]
$$

$$
= \mathbb{E}_{\mathcal{D}\sim U(\mathbb{D}),S\sim\mathcal{D}^m}\left[(\mathcal{E}^*(S) - L_{\mathcal{D}}(A(S)))^2\right] \qquad \square
$$

## I   Technical Lemma for Inestimability

**Lemma I.1.** *Let $m \in \mathbb{N}$, let $\varepsilon > 0$, let $\mathcal{X}$ be a finite set, let $\mathcal{F} \subseteq \{\pm 1\}^{\mathcal{X}}$ such that $\mathcal{F} \in \perp_{\varepsilon,\mathcal{X}}$, and let $A : (\mathcal{X} \times \{\pm 1\})^m \to \{\pm 1\}^{\mathcal{X}}$ be an $\mathcal{F}$-interpolating learning rule. For each $f \in \mathcal{F}$ let $\mathcal{D}_f = U(\{(x, f(x)) : x \in \mathcal{X}\})$, and for each $k \in \mathbb{N}$ let*

$$
p_k = \mathbb{P}_{\substack{F\sim U(\mathcal{F}) \\ S\sim(\mathcal{D}_F)^m}}\left[|\mathcal{F}_S| = k\right].
$$

*Then for any estimator $\mathcal{E} : (\mathcal{X} \times \{\pm 1\})^m \to [0, 1]$ that may depend on $A$,*

$$
\mathbb{P}_{\substack{F\sim U(\mathcal{F}) \\ S\sim(\mathcal{D}_F)^m}}\left[\left|\mathcal{E}(S) - L_{\mathcal{D}_F}(A(S))\right| \geq \frac{1}{4} - \frac{\varepsilon}{4}\right] \geq \sum_{k\in\{2,\dots,|\mathcal{F}|\}} \frac{p_k}{k}.
$$

*Proof.* Consider the following experiment:

1. Sample a sequence of points $X = (X_1, \dots, X_m)$ independently and uniformly at random from $\mathcal{X}$.

2. Sample a function $F$ uniformly from $\mathcal{F}$, independently of $X$.

3. For each $i \in [m]$, let $Y_i = F(X_i)$, let $Y = (Y_1, \ldots, Y_m)$, and let $S = \left((X_1, Y_1), \ldots, (X_m, Y_m)\right)$.

Let $\mathcal{P}$ be the joint distribution of $(X, F, Y, S)$. Fix $k \in \{2, \ldots, |\mathcal{F}|\}$, and let

$$s = \left((x_1, y_1), \ldots, (x_m, y_m)\right) \in (\mathcal{X} \times \{\pm 1\})^m$$

with $x = (x_1, \ldots, x_m)$ and $y = (y_1, \ldots, y_m)$ such that $|\mathcal{F}_s| = k$. Denote $\mathcal{F}_s = \{f_1, \ldots, f_k\}$. Then for any $i, j \in [k]$, $i \neq j$,

$$\begin{aligned}
\mathcal{P}(S = s \mid F = f_i) &= \mathcal{P}(X = x \mid F = f_i) \\
&= \mathcal{P}(X = x \mid F = f_j) && (X \perp F) \\
&= \mathcal{P}(S = s \mid F = f_j).
\end{aligned} \tag{13}$$

So,

$$\begin{aligned}
\mathcal{P}(F = f_i \mid S = s) &= \frac{\mathcal{P}(S = s \mid F = f_i) \cdot \mathcal{P}(F = f_i)}{\mathcal{P}(S = s)} \\
&= \frac{\mathcal{P}(S = s \mid F = f_j) \cdot \mathcal{P}(F = f_j)}{\mathcal{P}(S = s)} && (\text{By Eq. (13)}, F \sim \mathrm{U}(\mathcal{F})) \\
&= \mathcal{P}(F = f_j \mid S = s),
\end{aligned} \tag{14}$$

Seeing as $\mathcal{P}(F \in \mathcal{F}_s \mid S = s) = 1$, this implies that for all $i \in [k]$, $\mathcal{P}(F = f_i \mid S = s) = 1/k$.

Because $A$ is $\mathcal{F}$-interpolating, $A(s) \in \mathcal{F}_s$. Without loss of generality, denote $A(s) = f_1$. From $\mathcal{F} \in \perp_{\varepsilon, \mathcal{X}}$ and Fact 2.9, $L_{\mathcal{D}_{f_i}}(f_j) \geq \frac{1}{2} - \frac{\varepsilon}{2} := 2\alpha$ for all $i, j \in [k], i \neq j$. Hence,

$$\begin{aligned}
\mathcal{P}(L_{\mathcal{D}_F}(A(S)) = 0 \mid S = s) &= \mathcal{P}(F = A(S) \mid S = s) && (F, A(s) \in \mathcal{F}_s) \\
&= \mathcal{P}(F = f_1 \mid S = s) && (A(s) = f_1) \\
&= 1/k,
\end{aligned} \tag{15}$$

and

$$\begin{aligned}
\mathcal{P}(L_{\mathcal{D}_F}(A(S)) \geq 2\alpha \mid S = s) &= \mathcal{P}(F \neq A(S) \mid S = s) \\
&= \mathcal{P}(F \in \{f_2, \ldots, f_k\} \mid S = s) \\
&= (k-1)/k.
\end{aligned} \tag{16}$$

Hence, for any $\eta \in \mathbb{R}$,

$$\mathcal{P}\left(|L_{\mathcal{D}_F}(A(S)) - \eta| \geq \alpha \mid S = s\right) \geq \frac{1}{k}. \tag{17}$$

We conclude that for any estimator $\mathcal{E} : (\mathcal{X} \times \{\pm 1\})^m \to \mathbb{R}$,

$$\begin{aligned}
&\mathcal{P}(|L_{\mathcal{D}_F}(A(S)) - \mathcal{E}(S)| \geq \alpha) \\
&\quad \geq \sum_{k \in \{2, \ldots, |\mathcal{F}|\}} \mathcal{P}\left(|L_{\mathcal{D}_F}(A(S)) - \mathcal{E}(S)| \geq \alpha \bigwedge |\mathcal{F}_S| = k\right) \\
&\quad = \sum_{k \in \{2, \ldots, |\mathcal{F}|\}} \sum_{s : |\mathcal{F}_s| = k} \mathcal{P}\left(|L_{\mathcal{D}_F}(A(S)) - \mathcal{E}(S)| \geq \alpha \mid S = s\right) \cdot \mathcal{P}(S = s) \\
&\quad \geq \sum_{k \in \{2, \ldots, |\mathcal{F}|\}} \sum_{s : |\mathcal{F}_s| = k} \inf_{\eta \in \mathbb{R}} \mathcal{P}\left(|L_{\mathcal{D}_F}(A(S)) - \eta| \geq \alpha \mid S = s\right) \cdot \mathcal{P}(S = s) \\
&\quad \geq \sum_{k \in \{2, \ldots, |\mathcal{F}|\}} \sum_{s : |\mathcal{F}_s| = k} \frac{1}{k} \cdot \mathcal{P}(S = s) && (\text{By Eq. (17)}) \\
&\quad = \sum_{k \in \{2, \ldots, |\mathcal{F}|\}} \frac{1}{k} \cdot \mathcal{P}(|\mathcal{F}_S| = k)
\end{aligned}$$

as desired. $\qquad\square$

## J    Concentration Bound via Linear Programming

**Lemma J.1.** *Let $n \in \mathbb{N}$, $v_{\max} \in \mathbb{R}$. Let $Z$ be a random variable taking values in $[n]$ such that $\mu = \mathbb{E}[Z] \in [2, \sqrt{2}+1]$ and $\mathrm{Var}[Z] \leq v_{\max}$. Then $\mathbb{P}[Z \in \{2,3\}] \geq 1 - v_{\max}/2$.*

We prove this concentration of measure bound using the duality of linear programs (see Section 7.4.1 in Boyd and Vandenberghe, 2014 for an exposition of this approach).

*Proof.* Let $Z' = Z - \mu$. $Z'$ is a random variable with $\mathbb{E}[Z'] = 0$ and $\mathrm{Var}[Z'] = \mathrm{Var}[Z]$. Furthermore, $\mathbb{P}[Z \in \{2,3\}] = \mathbb{P}[Z' \in \{2-\mu, 3-\mu\}]$. We show a lower bound on $\mathbb{P}[Z' \in \{2-\mu, 3-\mu\}]$ across all distribution of $Z'$ with the above moment constraints.

Indeed, let $X$ be a random variable taking values in $\{1-\mu, 2-\mu, \ldots, n-\mu\}$ with $\mathbb{E}[X] = 0$ and $\mathrm{Var}[X] \leq v_{\max}$ such that $\mathbb{P}[X \in \{2-\mu, 3-\mu\}]$ is minimal. In particular, the distribution of $X$ is a solution to the following minimization problem.

$$\min_{\mathcal{D}_X} \mathbb{P}[X \in \{2-\mu, 3-\mu\}]$$

s.t.

$$\mathbb{E}[X] = 0$$
$$\mathrm{Var}[X] \leq v_{\max}$$

The minimization problem can be formulated as a linear program with variables $p_k = \mathbb{P}[X = k - \mu]$ for each $k \in [n]$.

$$\min_{\mathcal{D}_X} p_2 + p_3$$

s.t.

$$\sum_{k \in [n]} p_k \geq 1$$

$$\sum_{k \in [n]} -p_k \geq -1$$

$$\sum_{k \in [n]} p_k \cdot (k - \mu) \geq 0$$

$$\sum_{k \in [n]} p_k \cdot (\mu - k) \geq 0$$

$$\sum_{k \in [n]} -p_k \cdot (k - \mu)^2 \geq -v_{\max}$$

$$\forall k \in [n]: \ p_k \geq 0.$$

This linear program can be represented as

$$\min (0, 1, 1, 0, \ldots, 0) \cdot p$$

s.t.

$$\begin{pmatrix} 1 & 1 & \ldots & 1 \\ -1 & -1 & \ldots & -1 \\ 1-\mu & 2-\mu & \ldots & n-\mu \\ \mu-1 & \mu-2 & \ldots & \mu-n \\ -(1-\mu)^2 & -(2-\mu)^2 & \ldots & -(n-\mu)^2 \end{pmatrix} \begin{pmatrix} p_1 \\ \vdots \\ p_n \end{pmatrix} \geq \begin{pmatrix} 1 \\ -1 \\ 0 \\ 0 \\ -v_{\max} \end{pmatrix}$$

$$p \geq 0.$$

Recall the symmetric duality

$$\begin{array}{ccc} \min c^T x & & \max b^T y \\ \text{s.t.} & \longleftrightarrow & \text{s.t.} \\ \quad Ax \geq b & & \quad A^T y \leq c \\ \quad x \geq 0 & & \quad y \geq 0. \end{array}$$

Hence, the dual linear program is

$$\max (1, -1, 0, 0, -v_{\max}) \cdot y$$
s.t.

$$\begin{pmatrix} 1 & -1 & 1-\mu & \mu-1 & -(1-\mu)^2 \\ 1 & -1 & 2-\mu & \mu-2 & -(2-\mu)^2 \\ 1 & -1 & 3-\mu & \mu-3 & -(3-\mu)^2 \\ & & \vdots & & \\ 1 & -1 & n-\mu & \mu-n & -(n-\mu)^2 \end{pmatrix} \begin{pmatrix} y_1 \\ \vdots \\ y_5 \end{pmatrix} \leq \begin{pmatrix} 0 \\ 1 \\ 1 \\ 0 \\ \vdots \\ 0 \end{pmatrix}$$

$$y \geq 0.$$

A direct calculation shows that the vector

$$y^* = \left(1, 0, \alpha, 0, \tfrac{1}{2}\right), \qquad \alpha = \frac{1}{\mu-1} - \frac{\mu-1}{2}$$

is a feasible solution for the dual program for any $\mu \in [2, \sqrt{2}+1]$. The value of the dual program at $y^*$ is $u = 1 - v_{\max}/2$. The weak duality theorem for linear programs implies that $u$ is a lower bound on the value of the primal problem. Hence,

$$\min \mathbb{P}[X \in \{2-\mu, 3-\mu\}] \geq u.$$

This implies that $\mathbb{P}[Z \in \{2,3\}] \geq u$, as desired. $\qquad\square$

## K   Agreement Between Nearly-Orthogonal Functions

**Claim K.1.** *Let $\varepsilon > 0$, let $\mathcal{X}$ be a set, and let $f, g, h : \mathcal{X} \to \{\pm 1\}$ such that $\{f, g, h\} \in \perp_{\varepsilon, \mathcal{X}}$. Then $\mathbb{P}_{x \sim \mathrm{U}(\mathcal{X})}[f(x) = g(x) = h(x)] \leq \frac{1}{4} + \frac{3\varepsilon}{4}$.*

*Proof.* Denote

$$a = \mathbb{P}_{x \sim \mathrm{U}(\mathcal{X})}[f(x) = g(x) = h(x)]$$
$$b = \mathbb{P}_{x \sim \mathrm{U}(\mathcal{X})}[f(x) \neq g(x) = h(x)]$$
$$c = \mathbb{P}_{x \sim \mathrm{U}(\mathcal{X})}[f(x) = g(x) \neq h(x)]$$
$$d = \mathbb{P}_{x \sim \mathrm{U}(\mathcal{X})}[f(x) \neq g(x) \neq h(x)]$$

From $\{f, g, h\} \in \perp_{\varepsilon, \mathcal{X}}$ and Fact 2.9,

$$a + b = \mathbb{P}_{x \sim \mathrm{U}(\mathcal{X})}[g(x) = h(x)] \leq \frac{1}{2} + \frac{\varepsilon}{2}$$
$$a + c = \mathbb{P}_{x \sim \mathrm{U}(\mathcal{X})}[f(x) = g(x)] \leq \frac{1}{2} + \frac{\varepsilon}{2}$$
$$a + d = \mathbb{P}_{x \sim \mathrm{U}(\mathcal{X})}[f(x) = h(x)] \leq \frac{1}{2} + \frac{\varepsilon}{2}.$$

Adding these inequalities yields

$$3a + b + c + d \leq \frac{3}{2} + \frac{3\varepsilon}{2}.$$

From the identity $a + b + c + d = 1$,

$$2a \leq \frac{1}{2} + \frac{3\varepsilon}{2},$$

so $a \leq \frac{1}{4} + \frac{3\varepsilon}{4}$, as desired. $\qquad\square$

## L    Miscellaneous Lemmas

The following result from Achlioptas (2003) is a variant of a lemma of Johnson and Linden-strauss (1984).

**Theorem L.1** (Johnson–Lindenstrauss)**.** *Let $n, s \in \mathbb{N}$, let $\varepsilon, \beta > 0$, and let $V \subseteq \mathbb{R}^s$ be a set with cardinality $|V| = n$. Let $d \in \mathbb{N}$ such that*

$$d \geq \frac{4 + 2\beta}{\varepsilon^2/2 - \varepsilon^3/3} \ln(n).$$

*Let $R$ be a $d \times s$ random matrix such that each entry is chosen independently and uniformly at random from $\{\pm 1\}$. Let $f_R : \mathbb{R}^s \to \mathbb{R}^d$ be given by $f_R(v) = (1/\sqrt{d}) \cdot Rv$. Then*

$$\mathbb{P}_{R \sim \mathrm{U}(\{\pm 1\}^{d \times s})} \left[ \forall u, v \in V : \ (1 - \varepsilon) \|u - v\|_2^2 \leq \|f_R(u) - f_R(v)\|_2^2 \leq (1 + \varepsilon) \|u - v\|_2^2 \right] \geq 1 - \frac{1}{n^\beta}.$$

**Claim L.2** (Converse to Birthday Paradox)**.** *Let $d, m \in \mathbb{N}$, and let $\beta \in (0, 1)$. If*

$$m \leq \min \left\{ \sqrt{d \ln\left(\frac{1}{\beta}\right)}, \ \frac{d}{2} \right\}$$

*then $\mathbb{P}_{X \sim (\mathrm{U}([d]))^m}[|X| = m] \geq \beta$.*

*Proof.* We use the inequality $1 - x \geq e^{-x/(1-x)}$, which holds for $x < 1$.

$$\begin{aligned}
\mathbb{P}_{X \sim (\mathrm{U}([d]))^m}[|X| = m] &= 1 \cdot \left(1 - \frac{1}{d}\right) \cdot \left(1 - \frac{2}{d}\right) \cdots \left(1 - \frac{m-1}{d}\right) \\
&\geq \prod_{k=0}^{m-1} \exp\left(-\frac{k}{d-k}\right) = \exp\left(-\sum_{k=0}^{m-1} \frac{k}{d-k}\right) \\
&\overset{(*)}{\geq} \exp\left(-\frac{2}{d} \sum_{k=0}^{m-1} k\right) \geq \exp\left(-\frac{m^2}{d}\right),
\end{aligned}$$

where $(*)$ follows from $m \leq d/2$. Solving $\exp\left(-\frac{m^2}{d}\right) \geq \beta$ yields the desired bound.    □

**Theorem L.3** (Hoeffding, 1963)**.** *Let $a, b, \mu \in \mathbb{R}$ and $m \in \mathbb{N}$. Let $Z_1, \ldots, Z_m$ be a sequence of i.i.d. real-valued random variables and let $Z = \frac{1}{m} \sum_{i=1}^m Z_i$. Assume that $\mathbb{E}[Z] = \mu$, and for every $i \in [m]$, $\mathbb{P}[a \leq Z_i \leq b] = 1$. Then, for any $\varepsilon > 0$,*

$$\mathbb{P}[|Z - \mu| > \varepsilon] \leq 2 \exp\left(\frac{-2m\varepsilon^2}{(b-a)^2}\right).$$

## M    Experiments

### M.1    Motivation and Setup

Here, we examine if there are practical algorithms that admit loss stability or even hypothesis stability with substantial numerical values. To this end, we conduct experiments over a simple neural network architecture across four datasets: MNIST, FashionMNIST, CIFAR10, and CIFAR10 with random labels (figures 1-4, respectively). Throughout all experiments, we employ one-hidden-layer perceptrons with 512 hidden neurons. We train the models using stochastic gradient descent (SGD) with a momentum factor of 0.9 and a batch size of 1000, optimizing the cross-entropy loss. For every data set, we train the models across learning rates 0.1, 0.035,[22] and 0.01. We average all the curves over 10 random seeds (tied for the pairs of networks) and plot the standard deviation for all the curves.

---

[22]Except for CIFAR10, we present the results only for learning rate 0.1 and 0.01 to prevent clutter. The qualitative results are consistent across all datasets; that is, the curves of learning rate 0.035 lie between the curves of learning rate 0.1 and 0.01.

The training procedure is as follows: we train two models in tandem, starting from the same random initialization. The first model is provided with the full training set, whereas the second model has $k = 100$ data points removed from its training set. These points are drawn uniformly at random before the beginning of the training, and fixed thereafter. After each epoch, we evaluate the training accuracy, test accuracy and hypothesis stability, i.e., the agreement between the two models (which we calculate across the test set).

We set our main focus on the agreement of the models since the most amenable way to show loss stability might be by way of proving hypothesis stability. The latter can perhaps be mathematically proven in the case of neural networks by analyzing the stability of the training dynamics under two slightly different training sets.

### M.2 RESULTS

Across all experiments, the training and test accuracy of the model pairs are essentially identical throughout the training process. This suggests that at least simple models are loss stable across vision tasks. In order to reduce visual clutter, we hence only plot training and test accuracy of the first model (which has access to the full training set), respectively.

We observe higher agreement for simpler data sets and smaller learning rates. For example, the learning rate has a considerable effect on agreement for CIFAR10 ($\approx 0.65$ for learning rate 0.1 vs $\approx 0.8$ for learning rate 0.01).

The key takeaway from Figures 1 through 4 is that the agreement is consistently higher than the test accuracy. This relationship ensures that when applying the estimation procedure outlined in Theorem 4.3, we can avoid vacuous predictions of perfect accuracy. In the scenarios presented, the estimated accuracy will always be bounded away from 1, as it can be expressed as *test error + (1 - agreement)*. For instance, with a learning rate of 0.01, the maximum estimated accuracies are: 98% for MNIST (compared to 97.5% test accuracy), 90% for FashionMNIST (87% test accuracy), 72% for CIFAR10 (52% test accuracy), and 65% for CIFAR10 with random labels (10% test accuracy). These results illustrate a strong correlation between stability estimation and data complexity.

We repeat the same experiments, modifying the width of the hidden layer to investigate its impact on stability. The results, summarized in Table 1, reveal a strong positive correlation between network width and stability. This effect is particularly pronounced for more complex tasks, such as CIFAR10 and CIFAR10 with random labels. For instance, in the CIFAR10 random labels setting with a learning rate of 0.01, increasing the width from 256 to 1024 neurons improves agreement from 32% to 50%, highlighting the stabilizing effect of greater network width.

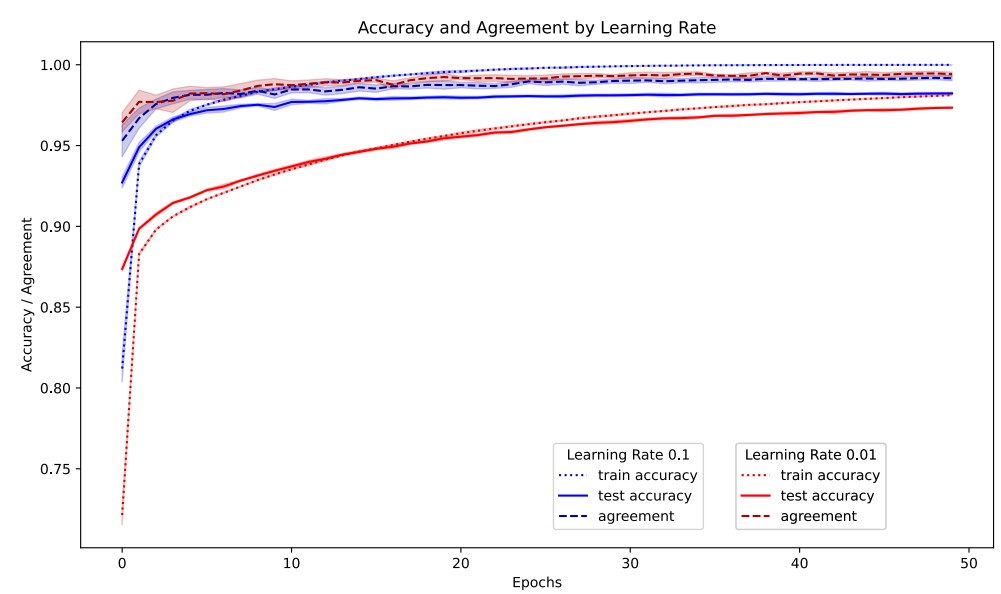

Figure 1: MNIST

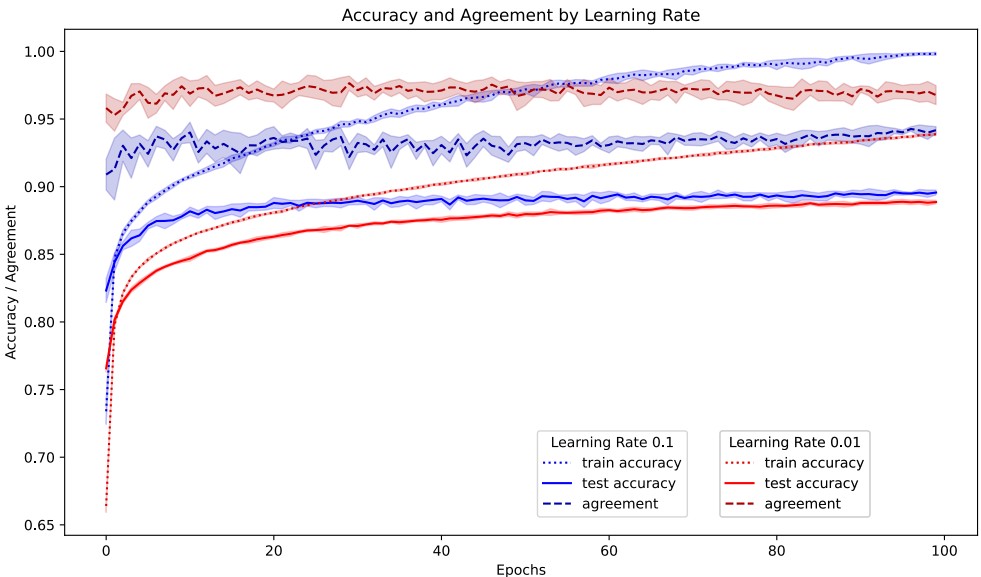

Figure 2: FashionMNIST

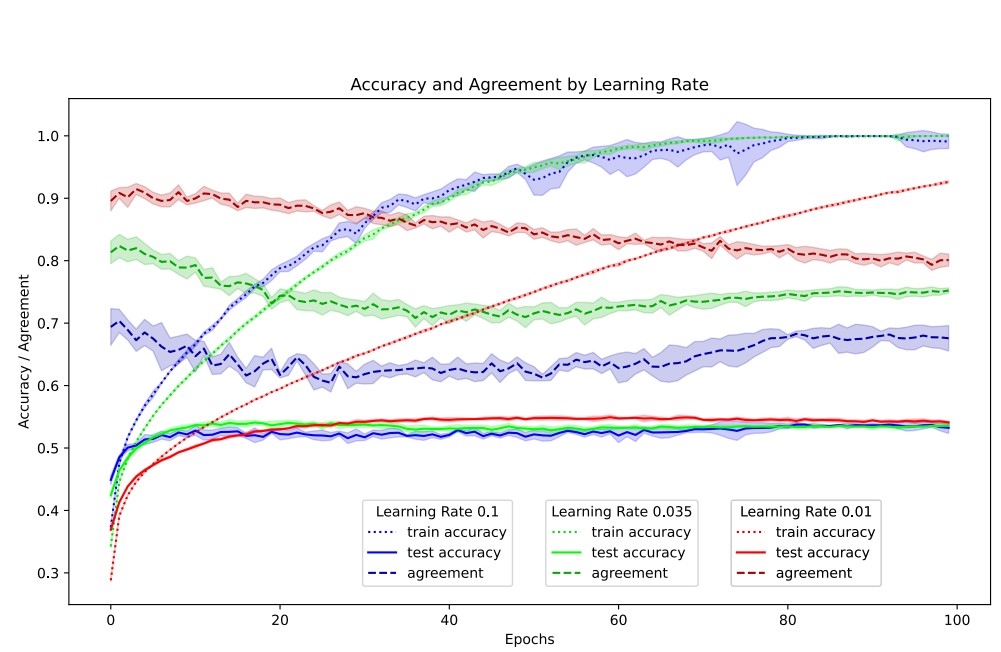

Figure 3: CIFAR10

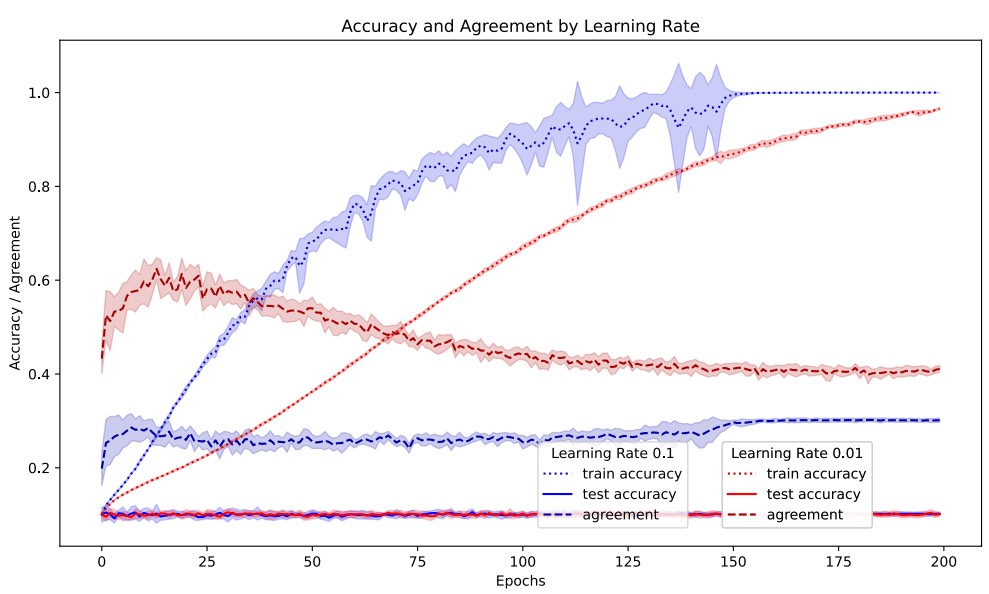

Figure 4: CIFAR10 with random labels

| MNIST | | | FMNIST | | | CIFAR10 | | | CIFAR10 - RAND | | |
|---|---|---|---|---|---|---|---|---|---|---|---|
| #N | lr | Agree | #N | lr | Agree | #N | lr | Agree | #N | lr | Agree |
| 256 | 0.1 | 99% | 256 | 0.1 | 92% | 256 | 0.1 | 62% | 256 | 0.1 | 21% |
| 256 | 0.01 | 99.5% | 256 | 0.01 | 97% | 256 | 0.01 | 71% | 256 | 0.01 | 32% |
| 512 | 0.1 | 99% | 512 | 0.1 | 94% | 512 | 0.1 | 67% | 512 | 0.1 | 30% |
| 512 | 0.01 | 99.5% | 512 | 0.01 | 97% | 512 | 0.01 | 80% | 512 | 0.01 | 41% |
| 1024 | 0.1 | 99% | 1024 | 0.1 | 95% | 1024 | 0.1 | 76% | 1024 | 0.1 | 39% |
| 1024 | 0.01 | 99.5% | 1024 | 0.01 | 98% | 1024 | 0.01 | 85% | 1024 | 0.01 | 50% |

Table 1: Agreement percentages across datasets with varying number of neurons in the hidden layer (#N) and learning rates (lr). The setup is the same as in M.1 except for the number of training epochs, which is $\{50, 150, 150, 300\}$ for {MNIST, FMNIST, CIFAR10, CIFAR10 random}, respectively. In scenarios where agreement has not yet reached saturation, agreement is positively correlated with the width of the network.

