# OpenReview forum: "Which Algorithms Have Tight Generalization Bounds?"
_ICLR.cc/2025/Conference — Submitted to ICLR 2025_

### Official Review · Reviewer_TgPZ · 2024-11-01

**Soundness:** 4
**Presentation:** 4
**Contribution:** 3
**Rating:** 8
**Confidence:** 3

**Summary:**

In prior work, it was established that it is not possible for generalization bounds to be tight uniformly across distributions if it does not explicitly incorporate the learning algorithm or data distribution. Building on this, the present paper studies conditions under which specific algorithms allow for tight generalization bounds, in the sense that they are close to the population loss for a family of distributions. Conditions are established for when this is possible (e.g., overparameterization and inductive bias) and when it is possible (a type of stability).

**Strengths:**

The contribution of the paper is significant in the sense that it formalizes and studies several notions related to generalization that are often discussed in loose terms. Taking a foundational approach like this to specify what goals we actually aim for, and conditions under which they are attainable, is important for the wider community. The presentation of the paper is overall clear and nicely written.

**Weaknesses:**

Some kind of small ablation in terms of model architecture could perhaps be interesting for the numerical experiments. At least something to indicate how the conclusions change with the scale of the model.

In some places, explicitly defining notation (even if it is standard) could be useful, e.g. $U(S)$ for uniform distribution on $S$.

**Questions:**

— This paper only considers deterministic algorithms. Are the general conclusions expected to hold for randomized ones? How would this differ?

— Can you comment on any connection to stability-focused works in the information-theoretic gen. bound literature, such as “Sample-Conditioned Hypothesis Stability Sharpens Information-Theoretic Generalization Bounds” by Wang and Mao? Is this aligned with the notions of stability you consider?

---

> ### Author Response · Authors · 2024-11-17
>
> >"Some kind of small ablation in terms of model architecture could perhaps be interesting for the numerical experiments. At least something to indicate how the conclusions change with the scale of the model."
>
> Thank you for suggesting this, we will add experiments that demonstrate how our notion of stability is affected by width. We will add networks with 256 and 1024 neurons. We ran some of the experiments already and what we observed is that the network becomes more stable as it gets wider.
>
> >"In some places, explicitly defining notation (even if it is standard) could be useful [...]"
>
> Thank you, we will make these additions.
>
> >"This paper only considers deterministic algorithms. Are the general conclusions expected to hold for randomized ones? How would this differ?"
>
> We agree that the topic of randomized learning algorithms is very important, seeing as most algorithms used in practice today are randomized. Indeed, in our research we did also investigate the case of randomized algorithms. However, we chose to focus on deterministic algorithms when we wrote this submission because it simplifies presentation (this paper is probably not a very easy read for some readers already, even without discussing randomized algorithms).
>
> To answer your question: yes, the estimability framework can be successfully extended to handle randomized algorithms as well, and in fact the original work [1] already contains some treatment of randomized algorithms.
>
> The main essence of the results we present in this paper carry over also to the case of randomized algorithms. This requires some technical work, but is definitely doable.
>
> The first step is to clearly define what estimability means for randomized learning algorithms. A definition that one might initially consider is that the estimator knows the randomness used by the algorithm, and must output a number that is with high probability close to the true population loss of the randomized algorithm. This definition is not very interesting, because a setting in which the estimator knows the randomness used by the randomized algorithm is equivalent to the setting of a deterministic algorithm, which we already handle. This is good to keep in mind though, because it means that our results for deterministic algorithms already apply as-is to randomized algorithms (like SGD) once the randomly chosen seed is fixed, which might be a simple and satisfactory approach for many purposes (SGD with a fixed random seed typically performs as well for most purposes as SGD with a fresh randomly-chosen seed).
>
> >"Can you comment on any connection to stability-focused works in the information-theoretic gen. bound literature, such as “Sample-Conditioned Hypothesis Stability Sharpens Information-Theoretic Generalization Bounds” by Wang and Mao? Is this aligned with the notions of stability you consider?"
>
> From our understanding, the work [2] by Wang and Mao adopts a stability definition similar in spirit to [3], using it to derive a generalization bound  (difference between true error and empirical error) through information-theoretic techniques. Our focus is on estimability—specifically, estimating the true error—rather than ensuring the stricter requirement that empirical error is close to the true error, as discussed in Lemma 7 of [3] and in [2]. Our notion of stability is easier to satisfy and captures a broader class of algorithms that should be considered stable. For instance, the memorization algorithm we mentioned in our paper in Example 1.7 admits an estimator that accurately predicts the true error, even though its empirical error is not close to the true error in many cases.
>
> **References:**
>
> [1] Gastpar, Michael, et al. "Fantastic Generalization Measures are Nowhere to be Found." The Twelfth International Conference on Learning Representations.
>
> [2] Wang, Ziqiao, and Yongyi Mao. "Sample-conditioned hypothesis stability sharpens information-theoretic generalization bounds." Advances in Neural Information Processing Systems 36 (2024).
>
> [3] Bousquet, O., & Elisseeff, A. (2002). Stability and generalization. The Journal of Machine Learning Research, 2, 499–526.

---

> > ### Comment · Reviewer_TgPZ · 2024-11-21
> >
> > Thank you for the thorough response and the additional experiments. I maintain my positive view of this submission.

---

> > > ### Author Response · Authors · 2024-11-25
> > >
> > > We sincerely appreciate your positive evaluation and support of our work. Thank you for recognizing the contributions of our study!
> > >
> > > As you suggested, we conducted additional experiments for widths 256 and 1024 in the hidden layer, see the table below (which we also now include in the newest revision in Appendix M.2).
> > >
> > >
> > > | **#N** | **lr** | **Agree** | **#N** | **lr** | **Agree** | **#N** | **lr** | **Agree** | **#N** | **lr** | **Agree** |
> > > |--------|--------|-----------|---------|--------|-----------|---------|--------|-----------|---------|--------|-----------|
> > > | **MNIST**       |            |           | **FMNIST**     |            |           | **CIFAR10**    |            |           | **CIFAR10 - RAND** |            |           |
> > > | 256    | 0.1    | 99%       | 256     | 0.1    | 92%       | 256     | 0.1    | 62%       | 256     | 0.1    | 21%       |
> > > | 256    | 0.01   | 99.5%     | 256     | 0.01   | 97%       | 256     | 0.01   | 71%       | 256     | 0.01   | 32%       |
> > > | 512    | 0.1    | 99%       | 512     | 0.1    | 94%       | 512     | 0.1    | 67%       | 512     | 0.1    | 30%       |
> > > | 512    | 0.01   | 99.5%     | 512     | 0.01   | 97%       | 512     | 0.01   | 80%       | 512     | 0.01   | 41%       |
> > > | 1024   | 0.1    | 99%       | 1024    | 0.1    | 95%       | 1024    | 0.1    | 76%       | 1024    | 0.1    | 39%       |
> > > | 1024   | 0.01   | 99.5%     | 1024    | 0.01   | 98%       | 1024    | 0.01   | 85%       | 1024    | 0.01   | 50%       |
> > >
> > >
> > >
> > >
> > > Our observation is that neural networks are becoming more stable the wider they are. This is actually quite interesting and adds more empirical insight, so thank you for suggesting to include these experiments!

---

### Official Review · Reviewer_rHm5 · 2024-11-02

**Soundness:** 4
**Presentation:** 3
**Contribution:** 3
**Rating:** 6
**Confidence:** 3

**Summary:**

This paper builds on the concept of the "Overparameterized setting" as defined in previous work (Gastpar et al., 2024), employing the recent notion of estimability to investigate conditions for the existence of tight generalization bounds for a given learning algorithm and distribution collection. The authors begin by showing that algorithms with certain inductive biases that lead to instability do not admit tight generalization bounds. Specifically, they demonstrate that when the amount of training data is significantly smaller than the hypothesis class’s VC dimension, it precludes worst-case estimability (Theorem 2.1).  Additionally, they establish inestimability for any algorithm with an inductive bias toward a class of nearly-orthogonal functions (Theorem 2.2).

Furthermore, they show that stability is a sufficient—but not necessary—condition for estimability. Notably, the paper's definition of stability differs from traditional leave-one-out algorithm stability (Bousquet and Elisseeff (2002)). They go on to prove that algorithms that are sufficiently stable do indeed possess tight generalization bounds (Theorem 3.3). Finally, they explain that whether an algorithm is estimable with respect to a set of distributions can be assessed by examining the conditional variance of the algorithm's loss, thereby relating the existence of tight generalization bounds to this conditional variance.

**Strengths:**

1. This paper addresses an often-overlooked issue in statistical learning theory: the concept of uniformly tight generalization bounds. By following the approach in Gastpar et al. (2024), the authors provide a particularly interesting perspective on the theoretical underpinnings of generalization error.

2. The detailed motivation and references regarding the definition of the overparameterized setting are invaluable for readers, especially as this overparameterized setting diverges from the more commonly understood notion of overparameterization (where the parameter count exceeds the sample size). Its purpose leans more towards serving as a definition for theoretical analysis.

3. The formal results and proofs appear sound and reasonably well-articulated, despite the fact that I have not verified every line of the proof.

4. Section 1.3 is particularly well-written, helping readers to quickly understand the paper’s theoretical results.

5. The paper’s approach of linking uniformly tight generalization bounds with estimability and traditional concepts like VC dimension and stability adds significant value, rendering the conclusions more intuitive and enhancing their relevance to established statistical learning theory.

**Weaknesses:**

1. The paper's title and abstract may give readers the impression that it explores which machine learning algorithms have tight generalization bounds in a broad context. However, the paper primarily focuses on the Overparameterized setting as defined within this work. While I acknowledge that the Overparameterized setting is prevalent in the deep learning community and holds significant theoretical value, there are models in deep learning that are not overparameterized, especially in resource-constrained scenarios or specific tasks, such as lightweight models designed to reduce parameter count and computational requirements. It would be beneficial for the authors to clarify in the abstract that their analysis is specific to the Overparameterized setting.

2. Adding more comparisons with Gastpar et al., 2024 could help to better delineate this paper’s contributions, as much of its setting and methodology are derived from that work. In fact, the central question of this paper—"Which machine learning algorithms have tight generalization bounds?"—originates from Question 2 in Gastpar et al., 2024: “For which algorithms does there exist a generalization bound that is tight for all population distributions in every overparameterized setting?” Therefore, a more detailed comparison with the conclusions of Gastpar et al., 2024, such as highlighting the additional insights provided by Theorem 2.1 in contrast to Theorem 2 in Gastpar et al., would enhance readers’ understanding of this paper’s contributions.

3. The statement in the related work section, “Our definition of an estimator EEE formalizes algorithm-dependent generalization bounds,” may be misleading, as the concept of estimability is derived from Gastpar et al., 2024, rather than being an original contribution of this paper.

4. The Introduction could benefit from a clearer summary of the main contributions. Additionally, the contributions of this paper appear somewhat limited, given that they build upon an already comprehensive theoretical framework established by Gastpar et al., 2024.

**Questions:**

The paper links the existence of tight generalization bounds to the conditional variance of the algorithm’s loss. How feasible is it to measure or approximate this conditional variance in practice, particularly for complex models like deep neural networks?

---

> ### Author Response · Authors · 2024-11-17
>
> > "It would be beneficial for the authors to clarify in the abstract that their analysis is specific to the Overparameterized setting."
>
> Thank you for pointing out this omission. In the shortly upcoming revised version, we will clearly state already in the abstract that our results apply specifically in the overparameterized setting. We remark that this is the only actually interesting regime. Estimability (of ERMs) is trivial in the not overparameterized setting (nr. of parameters./ VC dimension $\lesssim$ nr. of samples) since there the fundamental theorem of learning (see e.g. Thm. 6.8 in [1]) asserts that the empirical risk is a tight estimator of the population risk, for any ground truth distribution.
>
>
> > "Adding more comparisons with Gastpar et al., 2024 could help to better delineate this paper’s contributions, as much of its setting and methodology are derived from that work."
>
> Thank you for pointing this out. We will add a more detailed comparison to [2], similar to the one below, in the shortly upcoming revised version.
> To the best of our knowledge, our paper is the first to provide a rigorous and general mathematical formulation showing that any finite VC class admits inestimable algorithms (Theorem 2.1). This is somewhat surprising because it holds for all VC classes (which means that for any neural network architecture, there are some algorithms for which one will not be able to derive a tight generalization bound). We believe this is an important contribution. Furthermore, our results go well beyond those of [2]. The proofs of these results use various tools, including the JL lemma, our technical lemma (Lemma D.1), and the duality of linear programming. Note that the learnability–estimability tradeoff in Theorem 3 of [2] gives a limitation for algorithm-dependent bounds that is fairly abstract and involves a total variation (TV) condition that might be hard to check. In contrast, Theorems 2.1 and 2.2 involve very concrete combinatorial and geometric conditions (VC dimension, orthogonal functions). Even the more concrete result of [2] presented in Theorem 5, only holds for exactly orthogonal functions with strict algebraic structure (parity functions). In contrast, our Theorem 2.2 applies to generic $\epsilon$-orthogonal function classes (which contain exactly orthogonal functions as a special case).
> In contrast to [2], our work also presents positive results (Theorem 3.3 and Fact 4.2). Further, it is important to appreciate that Theorem 3.3 makes a nontrivial conceptual contribution by identifying the “correct” notion of stability for understanding estimability. The literature has many different notions of stability (see, e.g., [3]). Specifically, Reviewer 7dJN mentions algorithmic stability as in Lemma 7 of [4]. However, as we point out in line 290, that common definition of stability does **not** correspond to estimability. In particular, the memorization algorithm, which is very estimable, is not stable according to the definition of stability of [4]. We found this surprising! Although the memorization algorithm in Example 1.7 essentially always outputs the constant function $h(x)= -1$, it is not stable according to that definition. Therefore, we believe that Theorem 3.3 makes a meaningful contribution by identifying the specific notion of stability (Definition 3.2) that is sufficient for estimability, while holding for a broader class of algorithms compared to the stability notion of, e.g., [4].
>
> >"The statement in the related work section, “Our definition of an estimator E formalizes algorithm-dependent generalization bounds,” may be misleading, as the concept of estimability is derived from Gastpar et al., 2024, rather than being an original contribution of this paper."
>
> Thank you for catching this. We will fix this in the upcoming version.
>
> >"The Introduction could benefit from a clearer summary of the main contributions. Additionally, the contributions of this paper appear somewhat limited, given that they build upon an already comprehensive theoretical framework established by Gastpar et al., 2024."
>
> Please see our answer to your previous question concerning the comparison to [2]. To reiterate, we think that this paper makes important contributions that go well beyond the results of [2].
>
> **References:**
>
> [1] Shalev-Shwartz, Shai, and Shai Ben-David. Understanding machine learning: From theory to algorithms. Cambridge university press, 2014.
>
> [2] Gastpar, Michael, et al. "Fantastic Generalization Measures are Nowhere to be Found." The Twelfth International Conference on Learning Representations.
>
> [3] Moran, S., Schefler, H., & Shafer, J. (2023). The Bayesian Stability Zoo.
>
> [4] Bousquet, O., & Elisseeff, A. (2002). Stability and generalization. The Journal of Machine Learning Research, 2, 499–526.

---

> > ### Comment · Reviewer_rHm5 · 2024-11-26
> > **Thank You for the Response**
> >
> > Thank you to the author for the detailed response. I will maintain the current positive score, but will not assign a higher rating, as many of the settings and methods in this paper heavily draw from Gastpar et al., 2024.

---

### Official Review · Reviewer_D8Tm · 2024-11-05

**Soundness:** 3
**Presentation:** 2
**Contribution:** 3
**Rating:** 5
**Confidence:** 3

**Summary:**

The paper investigates the conditions under which machine learning algorithms exhibit tight generalization bounds through the concept of estimability (Definition 1.2). By first providing illustrative examples, the authors introduce estimability as a framework for understanding generalization. The paper establishes that certain inductive biases, which lead to algorithmic instability, can prevent the existence of uniformly tight generalization bounds. Conversely, stable algorithms are more likely to achieve tight bounds, highlighting stability as a favorable trait for reliable generalization. Finally, the paper introduces a novel characterization that connects tight generalization bounds to the conditional variance of an algorithm's loss.

**Strengths:**

The paper provides clear examples in Section 1.2 that illustrate the relationship between learnability, estimability, and good learning algorithms.

**Weaknesses:**

The paper is difficult to follow, primarily due to its reliance on concepts introduced in previous work. For example, footnote 3 does little to clarify the concept of overparameterization, and I found it necessary to refer back to Gastpar et al. (2024) to grasp these foundational ideas.

Additionally, immediately following Definition 1.10, the authors specify that only deterministic algorithms are considered in the paper. This limitation raises questions about the applicability of the theoretical framework to practical settings, where randomized algorithm often plays a role. Incorporating expectations across this “channel” of randomness, as done in information-theoretic bounds, would seem a complex but necessary extension to make the framework more practically relevant.

**Questions:**

Can the proposed framework be extended to handle randomized algorithms? If not, what challenges prevent this extension?

---

> ### Author Response · Authors · 2024-11-17
>
> >"The paper is difficult to follow, primarily due to its reliance on concepts introduced in previous work. For example, footnote 3 does little to clarify the concept of overparameterization, and I found it necessary to refer back to Gastpar et al. (2024) to grasp these foundational ideas."
>
> Thank you for your feedback. To make the paper self-contained and easier to follow, we will add a similar discussion as in Appendix D of [1] (https://openreview.net/forum?id=NkmJotfL42) in a new appendix section in our paper, together with the first paragraph of our footnote 3. The second paragraph of the footnote will go below Definition 1.4.

---

> ### Author Response · Authors · 2024-11-17
> **Cont.**
>
> >"Can the proposed framework be extended to handle randomized algorithms? If not, what challenges prevent this extension?
>
> We agree that the topic of randomized learning algorithms is very important, seeing as most algorithms used in practice today are randomized. Indeed, in our research we did also investigate the case of randomized algorithms. However, we chose to focus on deterministic algorithms when we wrote this submission because it simplifies presentation (this paper is probably not a very easy read for some readers already, even without discussing randomized algorithms)."
>
> To answer your question: yes, the estimability framework can be successfully extended to handle randomized algorithms as well, and in fact the original work [1] of Gastpar et al. (2023) already contains some treatment of randomized algorithms.
>
> The main essence of the results we present in this paper carry over also to the case of randomized algorithms. This requires some technical work, but is definitely doable.
>
> The first step is to clearly define what estimability means for randomized learning algorithms. A definition that one might initially consider is that the estimator knows the randomness used by the algorithm, and must output a number that is with high probability close to the true population loss of the randomized algorithm. This definition is not very interesting, because a setting in which the estimator knows the randomness used by the randomized algorithm is equivalent to the setting of a deterministic algorithm, which we already handle. This is good to keep in mind though, because it means that our results for deterministic algorithms already apply as-is to randomized algorithms (like SGD) once the randomly chosen seed is fixed, which might be a simple and satisfactory approach for many purposes (SGD with a fixed random seed typically performs as well for most purposes as SGD with a fresh randomly-chosen seed).
>
> >"[...] Incorporating expectations across this “channel” of randomness, as done in information-theoretic bounds, would seem a complex but necessary extension to make the framework more practically relevant."
>
> Perhaps the more interesting definition of estimability (compared to the one above) for randomized learning algorithms is one where the estimator knows the training set, but does not know the randomness used by the learning algorithm, and it is required to output a number that is close with high probability to the _expected_ population loss of the randomized algorithm when executed with this training set. (Perhaps this is what you meant by “Incorporating expectations across this ‘channel’ of randomness”?)
>
> In this setting, the essence of our results carry through, but there is one conceptual difference to keep in mind: using randomness, one can always engineer a learning algorithm that is estimable, essentially by adding noise to the output of the algorithm. As the noise in the algorithm’s output increases, the _expected_ 0-1 loss of the algorithm becomes closer to 1/2, and so the algorithm becomes estimable with a trivial estimator that simply always outputs the number 1/2. (With intermediate amounts of noise, a number between 0 and ½ will be optimal).
>
> Consequently, for randomized algorithms, our lower bounds in Theorems 2.1 and 2.2 can no longer be stated as absolute limitations on estimability. Rather there is now a tradeoff between the performance of the algorithm and its estimability. As we add more noise, the algorithm becomes more estimable, but its performance degrades. Thus, the theorems for randomized algorithms state that no algorithm can simultaneously make good predictions for a large set of labeling functions and also be estimable.
>
> On the other hand, the upper bound in Theorem 3.3 that states that stable algorithms are estimable remains basically unchanged for randomized algorithms.
>
> To summarize, one needs to be clear about the definition of estimability, and then our results do not change much, but the statement (and proof) of the lower bound are more complex. Does this answer your question regarding randomized algorithms?
>
> If it is important to you, we could attempt to add this treatment of randomized algorithms to the final version of the paper, to be submitted by Nov 27. Because this will require adding a bunch of technical and notational complexity, it might not actually make the paper better. It might be best to only add this as a remark or informal discussion. Let us know what you think!
>
> **References:**
>
> [1] Gastpar, M., Nachum, I., Shafer, J., & Weinberger, T. (2024). Fantastic generalization measures are nowhere to be found. The 12th International Conference on Learning Representations, ICLR 2024

---

> ### Author Response · Authors · 2024-11-27
> **Follow up**
>
> Dear Reviewer D8Tm,
>
> Thanks again for your thoughtful feedback on our paper. We have read your comments carefully and we have made some additions to the paper following your suggestions.
>
> In particular, we have added two appendices following your feedback:
>  * **Appendix B: On the Definition of Overparameterization.** We hope this appendix makes the paper more self-contained, and will make the notion of overparameterization used in the paper more accessible to our readers.
>  * **Appendix C: On Extending Our Results to Randomized Algorithms.** We hope this appendix answers your questions on estimability for randomized algorithms.
>
> At this point, we feel that we have addressed your feedback in full -- and we'd love to hear what you think!
>
> Best,
>
> The authors

---

### Official Review · Reviewer_7dJN · 2024-11-07

**Soundness:** 2
**Presentation:** 1
**Contribution:** 2
**Rating:** 1
**Confidence:** 4

**Summary:**

This work examines the conditions under which learning algorithms achieve tight bounds. To this end, the authors introduce a new definition of stability and explore its implications.

**Strengths:**

The paper is well-written.

**Weaknesses:**

- There are various mathematical models for over-parameterization in the literature. However, the authors do not mention related work on generalization error analysis within the over-parameterization regime. For instance, generalization error has been studied in frameworks such as the NTK [1] or Mean-field approaches [2,3,4]. In these studies, the authors do not assume zero training loss (or empirical risk) for all learning algorithms. Given that the primary claim of this paper concerns results for deep learning models, I believe the authors should compare their results to these works to better position their paper. Additionally, the authors refer to Appendix D of [6], which is incomplete.
- The current theoretical results are restrictive and apply only to binary classification.
- The experiments are insufficient and require further exploration. Moreover, some experimental details are missing. For instance, what are the loss functions used in the experiments?

**Questions:**

- Suppose someone extended the leave-one-out stability definition in [5] to a leave-k-out notion. What would be the specific difference between your stability notion and the leave-k-out notion?
- Could you provide a connection between your stability definition and the generalization error? For example, see Lemma 7 in [5].
- Why, in Definition 3.1, do you assume that $A$ is a mapping to $\\{\pm1\\}$?
- Why did the authors choose the neural network with 512 hidden neurons in the experiments? Why not 256 or fewer? Is there a particular reason?
- The authors define Definition 4.1 for the over-parameterized setting. Where is this used?

---

My current evaluation is strong reject. However, I am interested in reading authors's response.


---
**References:**

- [1]:Zixiang Chen, Yuan Cao, Quanquan Gu, and Tong Zhang. A generalized neural tangent kernel analysis for two-layer neural networks. Advances in Neural Information Processing Systems, 33: 13363–13373, 2020.
- [2]: Naoki Nishikawa, Taiji Suzuki, Atsushi Nitanda, and Denny Wu. Two-layer neural network on infinite dimensional data: global optimization guarantee in the mean-field regime. In Advances in Neural Information Processing Systems, 2022
- [3]: Atsushi Nitanda, Denny Wu, and Taiji Suzuki. Particle dual averaging: Optimization of mean field neural network with global convergence rate analysis. Advances in Neural Information Processing Systems, 34:19608–19621, 2021.
- [4]: Aminian, Gholamali, Samuel N. Cohen, and Łukasz Szpruch. "Mean-field Analysis of Generalization Errors." arXiv preprint arXiv:2306.11623 (2023).
- [5]: Bousquet, O., & Elisseeff, A. (2002). Stability and generalization. The Journal of Machine Learning Research, 2, 499–526.
- [6]: Gastpar, M., Nachum, I., Shafer, J., & Weinberger, T. (2024). Fantastic generalization measures are nowhere to be found. The 12th International Conference on Learning Representations, ICLR 2024.

---

> ### Author Response · Authors · 2024-11-17
>
> >“There are various mathematical models for over-parameterization in the literature. However, the authors do not mention related work on generalization error analysis within the over-parameterization regime. For instance, generalization error has been studied in frameworks such as the NTK [1] or Mean-field approaches [2,3,4]. In these studies, the authors do not assume zero training loss (or empirical risk) for all learning algorithms. Given that the primary claim of this paper concerns results for deep learning models, I believe the authors should compare their results to these works to better position their paper.”
>
>
> Thank you for your suggestion. We will add these references along with a brief discussion to better position our work. While the referenced papers study generalization bounds for specific types of algorithms, they do not establish whether these bounds are tight. Our work, in contrast, studies conditions that are necessary or sufficient for tight generalization bounds to exist in general, for any learning algorithm.
>
> As you pointed out, our paper concerns results for deep learning models. Neural networks typically **fit training data perfectly**, including random labels, making our focus on scenarios where the algorithm **achieves zero empirical error a strength**. In contrast, the works [1–4] consider algorithms that minimize empirical error + $\lambda \cdot D(q \| p)$, which may diverge from the behaviour of neural networks as they **do not guarantee zero empirical error—a limitation in this context**.
>
> To maintain clarity, we have opted not to extend our framework with additional notation for such algorithms. However, it is possible to combine insights from our work and [6] to show that any bound relying on a divergence term $D(q \| p)$, where $q$ is the true distribution and $p$ is a fixed prior, is vacuous in many scenarios. We will add this observation as a remark when discussing these papers.
>
> >“The current theoretical results are restrictive and apply only to binary classification.”
>
> The core of our technical contribution is the negative results for binary classification in Theorem 2.1 and Theorem 2.2. Many existing works–empirical and theoretical–in learning theory focus on binary classification, so this is not a huge limitation. In fact, the focus on binary classification makes our results stronger since we derive negative results! That is, since binary classification is a special case of multiclass classification (only two of $k$ classes have non-zero support in the data distribution) and regression (when the image of the loss function is discrete and binary), our negative results for binary classification directly imply negative results for multi-class classification and for regression.
> Additionally, our complementary positive results in Sections 3 and 4 readily hold for any bounded loss (e.g. they hold for multiclass classification and regression with bounded domain). We will clarify this in the paper.
>
> >“Additionally, the authors refer to Appendix D of [6], which is incomplete.”
>
> We referred to Appendix D found in the conference version, see (https://openreview.net/forum?id=NkmJotfL42). Perhaps you were looking at a different version? That appendix offers a comprehensive discussion about overparameterization and related quantities. If you think that parts of that discussion are incomplete, what specifically are you referring to? We would be happy to discuss.
>
> >“The experiments are insufficient and require further exploration. Moreover, some experimental details are missing. For instance, what are the loss functions used in the experiments?”
>
> The loss we used is the standard cross-entropy, thank you for noticing. Are there any other missing details?
>
> As Reviewer TgPZ suggested, we will add experiments that demonstrate how our notion of stability is affected by width. We will add networks with 256 and 1024 neurons. We ran some of the experiments already and what we observed is that the network becomes more stable as it gets wider. As our work is primarily theoretical, the experiments are intended to provide intuition rather than exhaustive empirical validation. Notably, the referenced theoretical papers [1] and [4] do not include experiments, and papers [2] and [3] contain fewer experiments than our paper, so we believe our work should be considered within the same context as these works.

---

> ### Author Response · Authors · 2024-11-17
> **Cont.**
>
> >“Suppose someone extended the leave-one-out stability definition in [5] to a leave-k-out notion. What would be the specific difference between your stability notion and the leave-k-out notion?”
>
> Thank you for your question. Extending the definition in [5] to a leave-$k$-out notion would not address the key limitation we highlight. The definition in [5], or the extension you suggest, requires that the predictions do not change much specifically across the removed samples versus our definition which requires that the output hypothesis does not change much over the whole domain when removing samples. To illustrate this, consider the memorization algorithm in Example 1.7. It is not stable according to [5] but it is stable according to our definition. Theorem 3.3 makes our definition a sufficient condition for estimability but since memorization is not stable according to [5] this makes the definition of [5] more restrictive than ours.
>
> >“Why, in Definition 3.1, do you assume that  A  is a mapping to {±1}?”
>
> Please note, in Definition 3.1, we do not assume that $A$ maps to {$\pm 1$}; rather, it maps samples of size $n$ to {$\pm 1$}$^\mathcal{X}$. So the range of $A$  is any binary function over the domain $\mathcal{X}$.
>
> >“Why did the authors choose the neural network with 512 hidden neurons in the experiments? Why not 256 or fewer? Is there a particular reason?”
>
> Thank you for your question. We chose a neural network with 512 hidden neurons as an educated guess to ensure it could fit random labels on CIFAR-10 (hardest testbed for stability), while, for example, 128 neurons are not sufficient. Additionally, the same number of neurons was used in the influential paper *Understanding Deep Learning Requires Rethinking Generalization* for fitting random labels, providing a well-established precedent for this choice.  Also, as we mentioned above, we will add experiments that explore the stability of SGD with 256 and 1024 neurons.
>
> >“The authors define Definition 4.1 for the over-parameterized setting. Where is this used?”
>
> Thank you for the comment. You are correct in that we do not explicitly use Definition 4.1 in the technical statements of the paper. However, the settings we consider in  Theorems 2.1 and 2.2  inherently imply overparameterization, such as a large VC-dimension or a large set of $\epsilon$-orthogonal functions, compared to the number of samples. Concretely, for Theorem 2.1 we consider an overparametrized setting that consists of all realizable distributions over a shattered set of size $d$ where the number of samples $m$ is less than $\sqrt{d}$; for Theorem 2.2  an overparametrized setting that consists of $2^{m}$   $(\Theta(1/m))$-orthogonal functions. Both cases are overparametrized settings because for neither of them there is an algorithm that can learn most of their distributions well - we use this fact in the proofs. We will add the above clarification as a remark in the paper.
>
> **References:**
>
> [1]:Zixiang Chen, Yuan Cao, Quanquan Gu, and Tong Zhang. A generalized neural tangent kernel analysis for two-layer neural networks. Advances in Neural Information Processing Systems, 33: 13363–13373, 2020.
>
> [2]: Naoki Nishikawa, Taiji Suzuki, Atsushi Nitanda, and Denny Wu. Two-layer neural network on infinite dimensional data: global optimization guarantee in the mean-field regime. In Advances in Neural Information Processing Systems, 2022
>
> [3]: Atsushi Nitanda, Denny Wu, and Taiji Suzuki. Particle dual averaging: Optimization of mean field neural network with global convergence rate analysis. Advances in Neural Information Processing Systems, 34:19608–19621, 2021.
>
> [4]: Aminian, Gholamali, Samuel N. Cohen, and Łukasz Szpruch. "Mean-field Analysis of Generalization Errors." arXiv preprint arXiv:2306.11623 (2023).
>
> [5]: Bousquet, O., & Elisseeff, A. (2002). Stability and generalization. The Journal of Machine Learning Research, 2, 499–526.
>
> [6]: Gastpar, M., Nachum, I., Shafer, J., & Weinberger, T. (2024). Fantastic generalization measures are nowhere to be found. The 12th International Conference on Learning Representations, ICLR 2024.

---

> > ### Author Response · Authors · 2024-11-25
> > **Looking forward to hearing from you**
> >
> > Dear Reviewer 7dJN,
> >
> > Thanks again for your thoughtful feedback on our paper. We have read your comments carefully and responded to them in full. We have also made some additions to the paper following your suggestions (including adding a discussion of NTK in Appendix A.1).
> >
> > If possible, we'd love to hear what you think now, before the discussion period ends.
> >
> > Best,
> >
> > The authors

---

> ### Author Response · Authors · 2024-12-02
> **Kindly reconsider your score?**
>
> Dear reviewer 7dJN,
>
> At the risk of being blunt, let us cut directly to the chase: We are very disappointed by the score you have assigned. We have read your feedback carefully, and it contains not a single substantial criticism of the actual mathematical or conceptual contributions of our paper (contributions that we maintain are strong). All the points mentioned in your review are small issues of presentation that are trivial to fix (see [summary table below](https://openreview.net/forum?id=RFMdtKbff5&noteId=82ZgsDJZtP)). Nonetheless, you have assigned a score of 1, which is typically reserved for “junk papers” that are fundamentally flawed. Seeing as you have not actually pointed out any meaningful flaws in our mathematical or conceptual contribution, we strongly urge you to reconsider your decision. At present it is badly uncalibrated.
>
> Generally, if a paper makes a meaningful contribution and does not have any significant weaknesses, we would respectfully suggest that a score of at least 6 ("weak accept") would be in order.
>
> Best,
>
> The authors
>
> $~$
>
> $~$
>
> ________________________
>
> $~$
>
> $~$
>
> > To enhance the presentation of your results, I recommend plotting both training and test losses.
>
> We already include train and test losses in Figures 1-4. In the final version of the paper, we will add similar graphs for the other experiments as well. (This is a minor issue that is trivial to fix.)
>
> $~$
>
> > Regarding the discussion of memorization in experiments, how can this phenomenon be effectively observed in neural networks? Could you provide an additional example of an algorithm that does not satisfies the stability definition by Bousquet and Elisseeff [2]?
>
> For example, consider how standard neural networks achieve 0 train error on random labels, while their test error on random labels is (of course) no better than random guessing (see, e.g., Figure 4 in our paper). This discrepancy between test and train errors shows that the algorithm is not stable according to the definition of Bousquet and Elisseeff [2], but it is quite stable according to our definition (as we show in Figure 4), and therefore is estimable.

---

> ### Author Response · Authors · 2024-12-02
> **Summary Table**
>
> The following table summarizes the reviewer's concerns and how we have addressed them:
>
> | **Topic** | **Significance** | **Response** |
> |-----------|-----------|--------------|
> | Comparison to specific overparametrized models such as NTK and mean field | Only tangentially related + minor issue | We care about a much more general class of overparametrized models; nevertheless, addressed in the new version per the request of the reviewer |
> | Results on binary classification | Not an issue | Negative results for binary classification imply negative results for multiclass and regression |
> | Definition 3.1 | Error on the reviewer side | Addressed in our response |
> | The loss function used in experiments | Minor point | Addressed in the new version |
> | Choice of network width | Minor point | Addressed in the new version, added new experiments with widths 256 and 1024 showing qualitatively similar behaviour |
> | Overparametrization | Used implicitly in the theorems | Addressed explicitly in the new version, see remark 3.3 |
> | Plotting both training and test losses | Already exists in the original version for 512 neurons and is an easy fix | We will add graphs for the other widths in the final version (256 and 1024) |
> | Relation between our stability, Bousquet et al. stability, generalization, and memorization algorithm | Addressed in the original version of the paper in sec. 4 | Further addressed in the new version following the reviewer 7dJN comments (sec. 1.4.2), neural network example (random labels) given in our above reply |

---

### Author Response · Authors · 2024-11-24
**New Version of the Paper Now Available!**

Dear reviewers,

Thanks again for your careful reading of our paper and your thoughtful comments!

Following your feedback, we have made a number of changes to the paper, including:

 * New **Related Works** subsection:
    - 1.4.1 Comparison to Gastpar et al. (2024)
    - 1.4.2 Stability
 * New **Appendices**:
    - A.1 Discussion of Neural Tangent Kernel and Mean-Field Theory
    - B On the Definition of Overparameterization
    - C On Extending Our Results to Randomized Algorithms
 * Addressed connections to **overparameterization**:
    - In new Remark 3.3.
    - In new footnote 6 and footnote 7 (on page 5).
 * Added some additional discussion in the **Our Results** section
    - "To the best of our knowledge, our paper is the first to..."
    - "Seeing as there are many definitions of stability in the literature..."
 * Added Remark 1.10, discussing **binary classification** vs. regression and multi-class classification
 * A number of smaller changes

We believe we have addressed essentially all your feedback, **except adding more experiments** (we are working on that, and will add them in the next few days).

We have uploaded the new version of the paper, and would love to hear what you think!

Best,

The authors

---

### Author Response · Authors · 2024-12-02
**Reaching a fair-minded final decision**

Dear Reviewers and Area Chairs,

It appears that the acceptance of this paper hinges on the score assigned by Review 7dJN.

That review does not find a single substantial weakness in the mathematical or conceptual contributions of our paper. All the points raised in that review are minor or tangential. Please see the [Summary Table](https://openreview.net/forum?id=RFMdtKbff5&noteId=82ZgsDJZtP) in that thread for a complete picture. Nonetheless, that review assigns a score of 1, which we believe is sorely miscalibrated.

We encourage everyone to take this into consideration when making final decisions.

Best,

The authors

---

### Meta-Review · Area_Chair_7ccC · 2024-12-22

**Metareview:**

This paper investigates which machine learning algorithms achieve tight generalization bounds in overparameterized settings, extending the findings of Gastpar et al. (2024). The authors identify conditions under which tight generalization bounds are unattainable, specifically for algorithms with inductive biases leading to instability. Conversely, they show that sufficiently stable algorithms can achieve tight bounds. The paper concludes with a simple characterization linking the existence of tight generalization bounds to the conditional variance of the algorithm's loss. The reviews for this paper are mixed, with scores ranging from 1 to 8. The main concerns raised by the reviewers are: (1) the paper is difficult to follow, primarily due to its reliance on concepts introduced in previous work, and (2) the authors did not address reviewer 7dJN’s comments properly. I believe the paper presents important contributions, but the rebuttal and author-reviewer discussions were not handled well. Given all these concerns, I recommend rejection.

**Additional Comments On Reviewer Discussion:**

During the rebuttal and author-reviewer discussion, Reviewer 7dJN found the authors' message (see below, which was later deleted), which was directed to another reviewer (Reviewer rHm5), unprofessional for dismissing a reviewer's comments by labeling them as an "outlier". I agree with Reviewer 7dJN's assessment.

Reviewer rHm5, who gave a rating of 6, stated, "I will maintain the current positive score, but will not assign a higher rating, as many of the settings and methods in this paper heavily draw from Gastpar et al., 2024."

Considering these points, I recommend rejection. The authors need to approach reviews with more professionalism, especially when responding to feedback that may be negative or disappointing.

---
Comment:
Dear Reviewer rHm5,
Thank you once again for your time and thoughtful reading of our paper.
We were very happy to receive a positive review from you. Normally, no further communication would be necessary at this point, however, in this case we feel we need to write to you again due to somewhat exceptional circumstances.

The issue is with review 7dJN. That review has not pointed out even a single substantial weakness in our paper, but nonetheless, it assigns our paper a score of 1. Please see the summary table in that thread -- the table clearly shows that all the comments in that review are concerned with minor or tangential points.
Seeing as review 7dJN is sorely miscalibrated, would you consider increasing your score to compensate? We feel that review 7dJN is an outlier, and it makes sense to correct for that outlier to get a more fair overall score for our paper.

Best,
The authors
---

---

> ### Public Comment · ~Jonathan_Shafer1 · 2025-02-20
>
> To clarify, the comment to reviewer rHm5 that was mentioned in the meta-review was written by me independently, without consulting my co-authors beforehand. In hindsight, I should have discussed it with them before posting on their behalf.
>
> I posted the comment on 2 Dec 2024, at 10:17 ET. Afterwards, my co-authors expressed reservations, so I removed it 36 minutes later (at 10:53 ET) with a note stating: "The below comment was deleted because it was not approved by all the authors."
>
> Any concerns regarding that comment should be addressed to me.
>
> That said, I respectfully maintain my personal opinion that the rating of 1 out of 10 in Review 7dJN was not well-supported. That single negative review ultimately outweighed three considerably more positive ones. I believe that steps should have been taken to evaluate the merits of the claims in Review 7dJN and ensure a more balanced decision.
>
> There were other concerns regarding the review process as well. For instance, Reviewer D8Tm declined to participate in the discussion period, which is not consistent with a careful and thorough review process.
>
> Best,
>
> Jonathan Shafer
>
> shaferjo.com

---

### Decision · Program_Chairs · 2025-01-22

Reject